# LoongRL: Reinforcement Learning for Advanced Reasoning over Long Contexts

**Siyuan Wang**[*,1,2]    **Gaokai Zhang**[*,1,3]    **Li Lyna Zhang**[1‡]
**Ning Shang**[1]    **Fan Yang**[1]    **Dongyao Chen**[2]    **Mao Yang**[1]
[1]Microsoft Research Asia    [2]Shanghai Jiao Tong University    [3]Carnegie Mellon University

## Abstract

Reasoning over long contexts is essential for large language models. While reinforcement learning (RL) enhances short-context reasoning by inducing "Aha" moments in chain-of-thought, the advanced thinking patterns required for long-context reasoning remain largely unexplored, and high-difficulty RL data are scarce. In this paper, we introduce **LoongRL**, a data-driven RL method for advanced long-context reasoning. Central to LoongRL is *KeyChain*, a synthesis approach that transforms short multi-hop QA into *high-difficulty* long-context tasks by inserting UUID chains that hide the true question among large collections of distracting documents. Solving these tasks requires the model to trace the correct chain step-by-step, identify the true question, retrieve relevant facts and reason over them to answer correctly. RL training on KeyChain data induces an emergent **plan–retrieve–reason–recheck** reasoning pattern that generalizes far beyond training length. Models trained at 16K effectively solve 128K tasks without prohibitive full-length RL rollout costs. On Qwen2.5-7B and 14B, LoongRL substantially improves long-context multi-hop QA accuracy by +23.5% and +21.1% absolute gains. The resulting LoongRL-14B reaches a score of 74.2, rivaling much larger frontier models such as o3-mini (74.5) and DeepSeek-R1 (74.9). It also improves long-context retrieval, passes all 128K needle-in-a-haystack stress tests, and preserves short-context reasoning capabilities. Code is available at https://loongrl.github.io/.

## 1 Introduction

Reasoning over long input contexts is a critical capability for large language models (LLMs), as many real-world tasks, from analyzing legal documents to debugging large codebases, require integrating information across thousands of tokens. Recent advances, such as OpenAI o-series (Jaech et al., 2024) and DeepSeek-R1 (Guo et al., 2025), show that reinforcement learning can improve reasoning by eliciting longer chain of thoughts (CoT) and emergent self-reflection. However, these methods mainly target short-context inputs and rely on model internal knowledge (e.g., math reasoning). In contrast, long-context reasoning requires both reasoning and the ability to retrieve and ground information from extensive external input contexts. Although modern models support longer context windows (Achiam et al., 2023; Ding et al., 2024), they excel mainly at retrieval, leaving reasoning over long documents a persistent challenge for real-world tasks (Ling et al., 2025).

This work aims to bridge this gap by enabling long-context models to move beyond retrieval and acquire advanced reasoning capabilities. Inspired by the short-context reasoning successes (Guo et al., 2025; Gandhi et al., 2025), we hypothesize that the key lies in discovering and mastering thinking patterns specific to long-context reasoning. Since such patterns remain unclear, we adopt a reinforcement learning approach to investigate whether high-quality reasoning patterns can emerge.

However, we face significant challenges. First, effective RL training requires difficult long-context problems that cannot be solved by retrieval alone. Questions must be sufficiently challenging to trigger reasonining and require retrieving relevant information from long input contexts. Moreover, since recent RL methods rely on outcome-only rewards to mitigate reward hacking (Liu et al., 2024;

---

[*]Equal contribution; work was done during the internship at Microsoft Research Asia
[‡]Correspondence to Li Lyna Zhang (li.zhang@xingyunzhili.com)

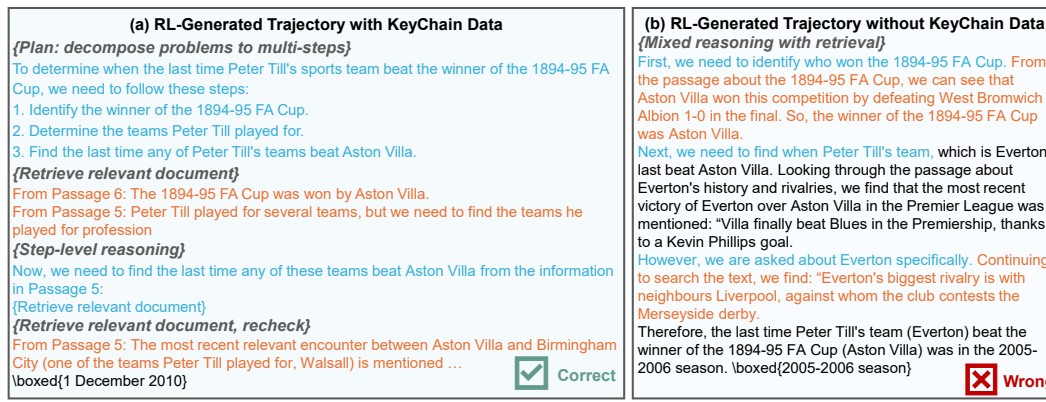

**(a) RL-Generated Trajectory with KeyChain Data**

*{Plan: decompose problems to multi-steps}*

To determine when the last time Peter Till's sports team beat the winner of the 1894-95 FA Cup, we need to follow these steps:
1. Identify the winner of the 1894-95 FA Cup.
2. Determine the teams Peter Till played for.
3. Find the last time any of Peter Till's teams beat Aston Villa.

*{Retrieve relevant document}*

From Passage 6: The 1894-95 FA Cup was won by Aston Villa.
From Passage 5: Peter Till played for several teams, but we need to find the teams he played for profession

*{Step-level reasoning}*

Now, we need to find the last time any of these teams beat Aston Villa from the information in Passage 5:

*{Retrieve relevant document}*

*{Retrieve relevant document, recheck}*

From Passage 5: The most recent relevant encounter between Aston Villa and Birmingham City (one of the teams Peter Till played for, Walsall) is mentioned …
\boxed{1 December 2010}  ✅ **Correct**

**(b) RL-Generated Trajectory without KeyChain Data**

*{Mixed reasoning with retrieval}*

First, we need to identify who won the 1894-95 FA Cup. From the passage about the 1894-95 FA Cup, we can see that Aston Villa won this competition by defeating West Bromwich Albion 1-0 in the final. So, the winner of the 1894-95 FA Cup was Aston Villa.

Next, we need to find when Peter Till's team, which is Everton, last beat Aston Villa. Looking through the passage about Everton's history and rivalries, we find that the most recent victory of Everton over Aston Villa in the Premier League was mentioned: "Villa finally beat Blues in the Premiership, thanks to a Kevin Phillips goal.

However, we are asked about Everton specifically. Continuing to search the text, we find: "Everton's biggest rivalry is with neighbours Liverpool, against whom the club contests the Merseyside derby.

Therefore, the last time Peter Till's team (Everton) beat the winner of the 1894-95 FA Cup (Aston Villa) was in the 2005-2006 season. \boxed{2005-2006 season}  ❌ **Wrong**

Figure 1: Model trajectories on long-context multi-hop QA with and without KeyChain RL data. **(a)** With KeyChain data, model exhibits an emergent plan–retrieve–reason–recheck thinking pattern, improving reasoning reliability and can generalize to longer contexts. **(b)** Without KeyChain data, reasoning and retrieval are entangled, the model often lacks an explicit planning step and does not deeply reason over retrieved information, frequently leading to errors. Reasoning steps are marked in blue and retrieval steps in orange.

Li et al., 2025; Shang et al., 2025a), the answers must be verifiable. In practice, however, such high-difficulty long-context questions are extremely scarce, and the ground-truth answers often take multiple valid forms, making reliable verification difficult. Second, strong long-context performance typically requires training at near-target lengths (Liu et al., 2024; Li et al., 2025), but scaling RL rollouts from short inputs (i.e., current <1K tokens) to 128K contexts incurs prohibitive compute and memory costs, making direct training infeasible at standard compute scales. Third, even if feasible, training exclusively on long-context data risks degrading short-context and general reasoning abilities (Peng et al., 2023; Shang et al., 2025b), which remain critical in practice.

To this end, we introduce **LoongRL**, a data-driven reinforcement learning method that incentivizes models to acquire effective thinking patterns for advanced long-context reasoning. At its core is **KeyChain**, a data synthesis approach that transforms short multi-hop QA datasets into *high-difficulty* long-context problems by extending contexts with distracting documents and inserting random UUID "chains" that hide the true question across multiple hops. These non-semantic UUID chains prevent the model from relying on lexical or semantic shortcuts during prediction, forcing it to trace the correct chains step-by-step, identify the actual question, retrieve relevant facts from the long context, and reason over them to generate the answer. Then, to enable effective RL training, we design a rule-based answer verifier, *two-way substring exact match*, which effectively evaluates free-form answers in general QA while mitigating reward hacking. Using KeyChain data, RL consistently elicits an emergent **plan–retrieve–reason–recheck** reasoning pattern, as shown in Fig. 1(a). Remarkably, this emergent patterns generalizes beyond the training length, enabling models trained at 16K to effectively handle 128K reasoning tasks without the prohibitive cost of full-length RL. Finally, we introduce a balanced data-mixing strategy to enhance long-context reasoning while preserving short-context general reasoning and long-context retrieval capabilities.

Extensive experiments across Qwen2.5-7B-Instruct and Qwen2.5-14B-Instruct and diverse benchmarks demonstrate the superiority of LoongRL. Remarkably, LoongRL substantially boosts Qwen2.5-7B-Instruct and Qwen2.5-14B-Instruct by **+23.5%** and **+21.1%** absolute accuracy improvements on long-context multi-hop QA tasks. The resulting LoongRL 14B achieves a score of 74.2, significantly surpassing all baselines and closely approaching much larger models such as o3 mini at 74.5 and DeepSeek-R1 at 74.9. Beyond the 16K training length, LoongRL generalizes effectively to inference lengths up to 128K tokens, substantially improving long-context retrieval and passing all needle-in-a-haystack pressure tests. At the same time, it preserves short-context and general reasoning capabilities, setting a new state of the art for models at this scale, and shows that LoongRL can induce advanced reasoning patterns to substantially improve long-context reasoning.

## 2 RELATED WORKS

**Reasoning and Long-Context Reasoning**. Recent advances in LLM reasoning are largely driven by high-quality human-like chains of thought (CoT), typically obtained via teacher model distilla-

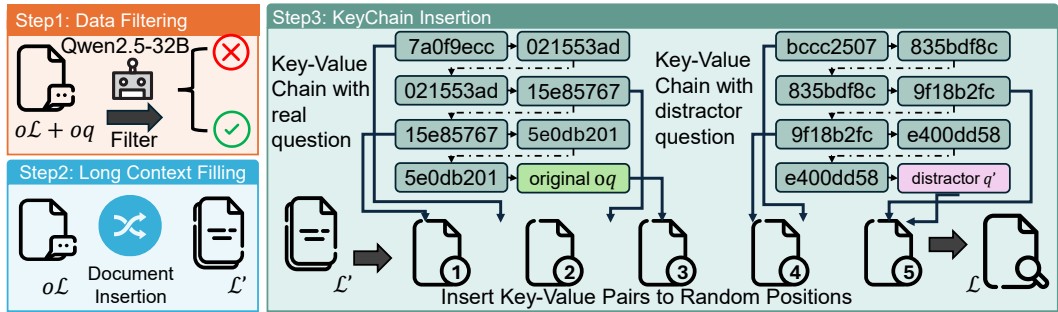

Figure 2: Overview of our KeyChain data construction.

tion (Yang et al., 2025) or self-generation through reinforcement learning (Guo et al., 2025). Most existing studies focus on short-context reasoning tasks, such as mathematics (Wang et al., 2025b; Shang et al., 2025a) and code (Liu et al., 2025; Ahmad et al., 2025), where emergent patterns like self-reflection and "aha" moments are crucial to the success (Gandhi et al., 2025). In contrast, exploration of advanced long-context reasoning patterns remains limited.

Existing efforts to improve long-context reasoning largely fall into two categories: prompting-based methods (Yen et al., 2024) and synthetic-data SFT (Li et al., 2024b;a;c). Prompting is limited by the base model's reasoning capacity, while synthetic-data SFT often introduces noise or bias, constraining advanced capabilities. QwenLong-L1 (Wan et al., 2025) makes a notable step by extending R1-distill-Qwen-32B with RL on sequences up to 60K tokens, encouraging self-exploration of long reasoning trajectories. However, it leaves open key questions about how to design high-quality RL training data. We address this gap by introducing KeyChain RL data that fosters emergent reasoning patterns and generalizes from 16K to 128K contexts with significantly higher efficiency.

**Long-Context Synthetic Data**. Existing methods for long-context data synthesis primarily extend input contexts by padding questions with additional irrelevant documents. For example, Li et al. (2024c) augment MuSiQue (Trivedi et al., 2022) with extra unrelated passages; Li et al. (2024a) use document-filling on HotpotQA (Yang et al., 2018) and SQuAD (Rajpurkar et al., 2016); and Li et al. (2024b) shuffle MuSiQue passages similarly. While these approaches increase context length, they are limited in generating high-quality, challenging training data.

## 3 METHODOLOGY

### 3.1 KEYCHAIN DATA CONSTRUCTION

**Overview**. LoongRL is a data-driven reinforcement learning approach designed to train models with advanced reasoning over long contexts. It relies on a high-quality RL training dataset $\mathbb{D} = \{\mathcal{L}_i, q_i, a_i\}$ constructed under three principles: **(i)** each question $q_i$ and answer $a_i$ are from real-world datasets to ensure reliability, as synthetic data often suffer from hallucination (Liu et al., 2025); **(ii)** solving question $q_i$ requires reasoning over the full long input context $\mathcal{L}_i$, not merely leveraging model internal knowledge or direct retrieval. **(iii)** questions $q_i$ are sufficiently challenging to allow RL to incentivize advanced long-context reasoning capabilities.

Fig. 2 illustrates the KeyChain data construction. We begin with curated, high-quality short-context QA pairs $\{o\mathcal{L}_i, oq_i, oa_i\}$ from real-world tasks. The original input context $o\mathcal{L}_i$ for each example is first expanded into a long input $\mathcal{L}'_i$ of 16K tokens by inserting distracting documents. KeyChain then transforms $\{\mathcal{L}'_i, oq_i, oa_i\}$ into $\{\mathcal{L}_i, q_i, a_i\}$ by randomly inserting multi-hop key-value chains that hide the original question $oq_i$ within $\mathcal{L}_i$, which significantly increases difficulty. Given the new question $q_i$, model must first traces the chain to recover $oq_i$, and then perform long-context reasoning over $\mathcal{L}_i$ to generate the correct answer $a_i$, where $a_i = oa_i$. This construction ensures that RL training focuses on reasoning over long contexts rather than memorization or shallow retrieval.

**Seed Dataset Curation and Context Extension**. We curate a high-quality seed dataset from three real-world multi-hop QA datasets: HotpotQA (Yang et al., 2018), MuSiQue (Trivedi et al., 2022), and 2WikiMultiHopQA (Ho et al., 2020). Each question $oq_i$ is paired with its ground-truth answer $oa_i$ and requires reasoning across multiple documents within a short context $o\mathcal{L}_i$. This initial col-

lection contains 277K QA instances. To ensure effective RL training, we filter out tasks that are overly easy or excessively hard questions. Specifically, we answer each question eight times using Qwen2.5-32B-Instruct (Qwen Team, 2024), and discard those with a pass rate of 0 or 1. This yields 72K examples of moderate difficulty.

We then extend each short context $o\mathcal{L}_i$ into a long context $\mathcal{L}'_i$ by inserting additional real-world documents while keeping the original question $oq_i$ unchanged. The extra documents are sampled from the short-context documents of the 200K filtered-out QA tasks, excluding any overlap with $o\mathcal{L}_i$. Each extended context is approximately $(<)$ 16,384 tokens, requiring the model to retrieve relevant information from a large set of distractors. This construction closely simulates real-world long-context reasoning, where relevant information is often buried within extensive irrelevant text.

**KeyChain Data Construction**. To make reinforcement learning effective for long-context reasoning, we build upon the above long-context multi-hop QA data to construct the KeyChain dataset. Fig. 2 illustrates the process. For each long-context QA task $\{\mathcal{L}'_i, oq_i, oa_i\}$, we insert linear key-value chains into the context, resulting in $\mathcal{L}_i$. In each chain, a key maps to a value that contains the next key, forming a step-by-step tracing path. We design two types of chains: *(i)* multiple chains that instead resolve to distracting questions, and *(ii)* one chain that ultimately resolves to the original question $oq_i$. Each key is generated as a 32-character UUID, with character randomly sampled from 0-9 and A-F.

From the distracting chains, we sample questions from other QA instances in the dataset, ensuring they are plausible but irrelevant. For target chain, we construct a new question $q_i$. This question requires the model to start from the initial key, trace the correct chain within $\mathcal{L}_i$, recover the original question $oq_i$, and finally perform long-context reasoning over $\mathcal{L}_i$ to produce the correct answer $a_i = oa_i$. This design substantially increases task difficulty, as the model must first localize the hidden question among multiple distractors and only then reason over the extended context to answer correctly. An example of an augmented KeyChain long-context question is shown below:

```
Example of KeyChain-augmented long-context question

Please read the following text.
<Document 0>
<original text> {"UUIDB-n":  "distracting question"} <original text>
<Document 1>
{"UUIDA-1":  "UUIDA-2"}
<Document 2>
{"UUIDB-1":  "UUIDB-2"}
...
{"UUIDA-n":  "correct question"}
...
In the context above, there is one correct question to answer.
The correct question can only be found by following the correct
consecutive chain of key:value pairs encoded with UUID strings
(e.g., f81d4fae-7dec-11d0-a765-00a0c91e6bf6), starting from
"starting UUIDA-1".
Find the correct question first, then answer it.
```

Importantly, our KeyChain method is generalizble and does not require a specific UUID format; the essential property is that the identifiers are high-entropy and non-semantic. This design prevents the model from exploiting lexical shortcuts during token generation and enforces explicit key–value retrieval. In our experiments, performance remains nearly identical when UUIDs are replaced with randomly generated strings, as shown in Appendix A.10.

**Emergent Long-Context Reasoning Patterns**. We surprisingly find that RL training with Key-Chain data enables models to develop emergent, human-like long-context reasoning patterns. As shown in Fig. 1(a), for each long-context QA task, the model first generates an explicit plan decomposing the problem into subproblems and substeps, retrieves relevant information for each step, and actively re-checks retrieved content when uncertain before proceeding. This structured **plan–retrieve–reason–recheck** loop leads to highly logical and reliable solutions. Furthermore, we observe that this reasoning pattern also improves conventional long-context retrieval tasks, as illustrated in Appendix A.5 with an example trajectory on the RULER vt benchmark, where the model

performs step-by-step, human-readable retrieval, progressively locating the correct answer rather than directly jumping to it as in traditional retrieval approaches.

More importantly, the plan–retrieve–reason–recheck behavior learned on short contexts (16K tokens) **generalizes to much longer contexts**, up to 128K tokens (see Experiments). This allows training on 16K sequences while maintaining strong longer context performance, highlighting the robustness and scalability of the KeyChain RL approach.

## 3.2 LONG-CONTEXT REINFORCEMENT LEARNING

This section introduces our long-context reinforcement learning methodology using KeyChain data, covering reward design, data mixing and multi-stage training recipe.

### 3.2.1 GROUP RELATIVE POLICY OPTIMIZATION FOR LONG-CONTEXT REASONING

**Group Relative Policy Optimization (GRPO)**. For training, we adopt the GRPO algorithm. Specifically, for each question $q$, its long context $\mathcal{L}$, and its ground-truth answer $a$ from a dataset $D$, GRPO samples a group of rollout trajectories $\{o_1, o_2, \cdots, o_G\}$ from the old policy $\pi_{\theta_{old}}$ and then optimizes the policy $\pi_\theta$ by maximizing the following objective:

$$J_{\text{GRPO}}(\theta) = \mathbb{E}_{(\mathcal{L},q,a)\sim\mathcal{D}, \{o_i\}_{i=1}^G \sim \pi_{\theta_{\text{old}}}(\cdot|q)}$$

$$\left[ \frac{1}{G} \sum_{i=1}^{G} \frac{1}{|o_i|} \sum_{t=1}^{|o_i|} \Big( \min\big[\rho_{i,t}(\theta)A_{i,t}, \text{clip}(\rho_{i,t}(\theta), 1-\varepsilon, 1+\varepsilon)A_{i,t}\big] \quad -\beta D_{\text{KL}}(\pi_\theta\|\pi_{\text{ref}})\Big) \right] \tag{1}$$

where $\rho_{i,t}(\theta) = \frac{\pi_\theta(o_{i,t}|q,o_{i,<t})}{\pi_{\theta_{\text{old}}}(o_{i,t}|q,o_{i,<t})}$. Hyper-parameters $\varepsilon$ and $\beta$ control the clipping range of importance sampling ratio and the weight of KL penalty term, respectively. The estimated advantage $A_{i,t}$ is computed from a group of rewards $\{r_1, r_2, ...r_G\}$ for each rollout trajectory:

$$A_{i,t} = \frac{r_i - \text{mean}(\{r_1, r_2, \cdots, r_G\})}{\text{std}(\{r_1, r_2, \cdots, r_G\})} \tag{2}$$

Here, $r_i$ is the reward for trajectory $o_i$, evaluated using a rule-based verifier to mitigate reward hacking (Guo et al., 2025; Kimi Team et al., 2025).

To stabilize RL training, we follow best practices. A small KL penalty $\beta = 0.001$ prevents excessive policy deviation. Following prior works (Shang et al., 2025a), we remove the entropy loss term, which while commonly used to encourage exploration, can cause uncontrolled entropy growth and destabilize training, so it is omitted in our experiments.

**Rule-based Reward Design**. In our long-context RL, most questions are general QA rather than math or code problems with clear answers. The answers can take many valid forms, making it difficult to determine whether a rollout trajectory truly reaches the correct solution. Prior works such as QwenLong-L1 (Wan et al., 2025) address this by using LLM-as-a-judge, but this introduces additional complexity. In addition to the already expensive long-context RL training, it requires serving another model for answer judgment, while still leaving room for reward hacking.

We instead adopt a rule-based reward, following the success of verifiable rewards in mathematical and code RL (Shang et al., 2025a; Wang et al., 2025a; Guo et al., 2025). Our approach is simple yet effective for long-context reasoning. First, we explicitly require the model to output its final answer within `\boxed{}` in the training prompt (in Appendix. A.2), ensuring unambiguous answer extraction. Second, we apply a *two-way substring exact match* on the boxed answer. Each rollout trajectory $o_i$ receives a binary accuracy reward $r_i \in \{0, 1\}$ depending on whether the extracted final answer $y_{\text{ans}}$ contains the ground truth answer $a$ as a substring, or the ground truth answer $a$ contains $y_{\text{ans}}$ as a substring. Formally, the reward is computed as:

$$r_i = \begin{cases} 1 & \text{if } \{a \subseteq y_{\text{ans}} \ \lor \ y_{\text{ans}} \subseteq a\}, \\ 0 & \text{otherwise.} \end{cases} \tag{3}$$

Compared to strict exact match, this design tolerates valid answer variations and avoids the rigidity that may otherwise exclude correct outputs. Experiments in Table 6 demonstrate its effectiveness.

Table 1: Data recipe for long-context RL training.

| Dataset | Description | # Size | Length range | Difficulty |
|---|---|---|---|---|
| HotpotQA-KeyChain | KeyChain-augmented HotpotQA | 2,500 | 16,272–20,670 | Hard |
| MuSiQue-KeyChain | KeyChain-augmented MuSiQue | 2,500 | 16,495–20,623 | Hard |
| 2WikiMultiHopQA-KeyChain | KeyChain-augmented 2WikiMultiHopQA | 2,500 | 14,911–20,576 | Hard |
| HotpotQA | Standard multi-hop QA | 2,500 | 12,058–16,279 | Medium |
| MuSiQue | Standard multi-hop QA | 2,500 | 12,562–16,283 | Medium |
| 2WikiMultiHopQA | Standard multi-hop QA | 2,500 | 10,727–16,274 | Medium |
| Book RULER (Multi-key) | Long-context retrieval (20 keys, 1 value) | 512 | 12,038–17,387 | Easy |
| Book RULER (Multi-value) | Long-context retrieval (1 key, 20 values) | 512 | 11,648–17,840 | Hard |
| Math Choice | Multiple-choice math problems | 2,500 | 40–425 | Easy |
| DAPO Math | Mathematical reasoning | 2,500 | 65–1,014 | Hard |

### 3.2.2 TRAINING RECIPE

We conduct LoongRL training on Qwen2.5-7B-Instruct and Qwen2.5-14B-Instruct, both with a 128K context window. The goals are (i) enhancing long-context reasoning through reinforcement learning, and (ii) preserving core abilities such as general short-context reasoning. To achieve this, we construct a mixed dataset (Table 1) and adopt a multi-stage RL training strategy.

**Training Length and Data Mix**. As discussed in Sec. 3.1, KeyChain data effectively induces long-context reasoning patterns, enabling the model to generalize to longer contexts. To avoid the high cost of full 128K RL rollouts, we train using a 16K context length.

Table 1 summarizes our training data sources, including their input context lengths and task difficulty. Our dataset consists of four types. *(i) High-difficulty KeyChain data* is synthesized as described in Section 3.1, with 2,500 examples each from HotpotQA (Yang et al., 2018), MuSiQue (Trivedi et al., 2022), and 2WikiMultiHopQA (Ho et al., 2020), totaling 7,500 examples. This set provides challenging examples that explicitly induce long-context reasoning. *(ii) Medium-level multi-hop QA data* consists of 2,500 examples from each of the same three datasets. These moderately difficult examples are especially important for smaller models (e.g., Qwen2.5-7B-Instruct), enabling effective RL when the model initially struggles with harder KeyChain tasks. *(iii) Long-context needle retrieval data* contains 1,024 synthetic examples designed to maintain the model's ability to retrieve relevant information from long contexts. Each example uses a 16K-token book from PG19 as the base, into which multiple key–value "needles" are randomly inserted following RULER(Hsieh et al., 2024), requiring the model to locate relevant values amid extensive distractors. *(iv) Math data* contains 5,000 short-context problems ($<$1K tokens) to preserve general short-context reasoning capabilities, including 2,500 hard problems from the DAPO training set (Yu et al., 2025) and 2,500 easy multiple-choice questions from MATH (Hendrycks et al., 2021).

**Multi-Stage Training**. Our reinforcement learning follows a three-stage curriculum. *(i) Warm-up*. We first train for one epoch on the dataset excluding KeyChain data. Since KeyChain problems are initially too difficult for small models, this stage allows the model to improve retrieval and general reasoning ability on easier data, ensuring stable optimization. *(ii) Stage I (KeyChain augmentation)*. KeyChain data is then introduced to increase task difficulty, encouraging the model to plan effectively, retrieve precise information from distractor-heavy long contexts, and integrate evidence into coherent reasoning chains. *(iii) Stage II (difficulty-focused training)*. After Stage I, we generate eight rollouts per example using the best checkpoint. Examples solved correctly in all rollouts are discarded, leaving a challenging subset ( 30–40% of the data). RL continues on this subset, focusing updates on difficult cases to improve efficiency while avoiding overtraining on mastered problems.

## 4 EXPERIMENTS

### 4.1 SETUP

**Training Setup.** We run experiments on two long-context instruction-tuned models, *Qwen2.5-7B-Instruct* and *Qwen2.5-14B-Instruct*. Training uses GRPO using a group size $G = 8$ and a learning rate of 1e-6. Batch sizes are set to 512 for 7B model and 256 for 14B model. Rollouts are sampled with temperature 0.6 and top-$p = 0.95$, with a maximum output length of 4,096 tokens and long-context inputs of $\sim$ 16K. We adopt a learning rate of $1 \times 10^{-6}$ with cosine decay and gradient

Table 2: Results of LoongRL and frontier LLMs on long-context reasoning and general short tasks. LoongRL delivers frontier-level long-context reasoning at much smaller scales (7B/14B), rivaling o3-mini and DeepSeek-R1, while preserving general short-context abilities across all scales.

| Models | Long-Context Reasoning | | | | | | General & Short Reasoning | | | |
|---|---|---|---|---|---|---|---|---|---|---|
| | Avg. | HotpotQA | 2WikiMultiHopQA | MuSiQue | NarrativeQA | QASPER | Avg. | MMLU | MATH | IFEval |
| o3–mini (medium) | 74.5 | 83.0 | 89.0 | 64.0 | 60.7 | 60.5 | **92.1** | 86.9 | **98.0** | **91.5** |
| DeepSeek- R1 | **74.9** | 82.7 | 91.3 | **72.2** | **66.9** | 61.4 | 90.5 | **90.8** | 97.3 | 83.3 |
| GPT–4o | 64.7 | 82.5 | 78.0 | 54.0 | 60.5 | 48.5 | 82.5 | 88.7 | 74.6 | 84.3 |
| QwQ-32B | 69.6 | 78.5 | 87.4 | 62.7 | 61.1 | 58.5 | 85.9 | 75.7 | **98.0** | 83.9 |
| R1-Distill-LLaMa-70B | 65.4 | 76.1 | 85.0 | 61.9 | 53.4 | 50.5 | 85.4 | 82.4 | 94.5 | 79.3 |
| Qwen2.5-7B-Instruct | 48.9 | 69.5 | 50.5 | 34.0 | 44.5 | 46.0 | 73.5 | 73.4 | 76.0 | **71.2** |
| R1-Distill-Qwen-7B | 31.2 | 40.2 | 53.3 | 11.1 | 8.9 | 42.5 | 69.9 | 62.3 | **92.8** | 54.7 |
| **LoongRL-7B** | **72.4** | 83.1 | 91.1 | 65.6 | 58.4 | 63.6 | **75.0** | 76.2 | 78.0 | 70.9 |
| Qwen2.5-14B-Instruct | 53.1 | 74.0 | 60.5 | 36.5 | 48.5 | 46.0 | 81.3 | 79.4 | 83.4 | **81.0** |
| R1-Distill-Qwen-14B | 64.9 | 77.5 | 87.0 | 58.0 | 51.0 | 51.0 | 81.0 | 76.6 | 93.9 | 72.6 |
| R1-Distill-Qwen-32B | 65.5 | 76.3 | 87.6 | 59.8 | 52.7 | 50.9 | 82.4 | 80.5 | 94.3 | 72.5 |
| QwenLong-L1-32B | 70.1 | 80.7 | 89.1 | 65.2 | 58.6 | 56.7 | **84.1** | 78.5 | **95.2** | 78.6 |
| **LoongRL-14B** | **74.2** | 82.2 | 93.3 | 67.5 | 63.4 | 64.5 | 80.7 | 80.5 | 83.2 | 78.4 |

Table 3: Results of LoongRL and frontier LLMs on the HELMET long-context generation benchmark. LoongRL-7B and LoongRL-14B substantially outperform baseline models.

| Model | Avg. | RAG | | | | Generation with Citations | | Summarization | |
|---|---|---|---|---|---|---|---|---|---|
| | | NaturalQuestions | HotpotQA | PopQA | TriviaQA | ALCE ASQA | ALCE Qampari | ∞Bench | Multi-LexSum |
| Qwen2.5-7B-Instruct | 22.8 | 21.2 | 20.3 | 27.7 | 53.3 | 6.1 | 0.1 | 14.7 | 39.2 |
| R1-Distill-Qwen-7B | 6.7 | 4.7 | 4.0 | 19.2 | 9.2 | 11.6 | 5.2 | 0 | 0 |
| **LoongRL-7B** | **44.8** | **50.0** | **60.0** | **53.7** | **82.8** | **15.0** | **7.8** | **28.1** | **60.8** |
| Qwen2.5-14B-Instruct | 40.5 | 40.5 | 48.0 | 45.3 | 81.7 | 24.1 | 6.6 | 27.7 | 50.4 |
| R1-Distill-Qwen-14B | 29.7 | 41.8 | 41.7 | 40.8 | 32.0 | 14.9 | 9.7 | 16.1 | 40.4 |
| QwenLong-L1-32B | 41.8 | 46.5 | 44.7 | 35.5 | 85.3 | 16.3 | 12.2 | 28.2 | **65.8** |
| **LoongRL-14B** | **49.0** | **55.3** | **60.3** | **51.0** | **88.2** | **23.0** | **14.3** | **33.7** | 65.8 |

clipping at 1.0. For Qwen2.5-7B-Instruct, we apply the full three-stage RL training: 42 steps in warm-up, 168 in Stage I and 118 in Stage II. For the larger Qwen2.5-14B-Instruct, we skip warm-up stage since the model already possesses strong base abilities and can immediately handle KeyChain data. We train for 168 steps in Stage I and 150 steps in Stage II. For the 7B model, we train on 16×A100 GPUs, while the 14B model is trained on 8×MI300X GPUs.

**Evaluation Benchmarks**. We evaluate LoongRL models across three dimensions. **(i) Long-context reasoning**: we follow QwenLong-L1 (Wan et al., 2025) and evaluate on multi-hop QA tasks in LongBench v1 (Bai et al., 2024), including HotpotQA (Yang et al., 2018), 2WikiMulti-HopQA (Ho et al., 2020), MuSiQue (Trivedi et al., 2022)), NarrativeQA (Kočiský et al., 2018) and QASPER (Dasigi et al., 2021), with input lengths from 4K to 64K tokens. We also evaluate evaluate on HELMET (Yen et al., 2024), a more challenging long-context generation benchmark. **(ii) General short-context reasoning**: we use standard benchmarks including MMLU (Hendrycks et al., 2020), MATH-500 (Lightman et al., 2023), and the instruction-following benchmark IFEval (Zhou et al., 2023). **(iii) Long-context retrieval**: to measure the impact of long-context RL on retrieval abilities, we evaluate on Needle in a Haystack (Kamradt, 2023) and RULER (Hsieh et al., 2024).

For inference, reasoning models and our models use temperature 0.6, with up to 128K input tokens and 10K output tokens. We sample eight solutions per problem and report average pass@1 accuracy. Non-reasoning models (e.g., Qwen2.5-7B-Instruct) use temperature 0.

**Baselines**. We compare against three baselines: **(i)** leading frontier models, including o3-mini, GPT-4o, DeepSeek-R1 and QWQ-32B; **(ii)** state-of-the-art models enhancing short-context reasoning on long-context foundations, mainly R1-distilled variants; and **(iii)** long-context reasoning models like the recent QwenLong-R1-32B, based on R1-distill-Qwen-32B and trained with 60K input context.

### 4.2 MAIN RESULTS

**Competitive long-context reasoning at smaller scale**. Table 2 summarizes the long-context reasoning performance of LoongRL against state-of-the-art models. We highlight two key observations: **(i)** LoongRL delivers frontier-level long-context reasoning at significantly smaller scales. Remarkably, LoongRL-7B achieves an average of 72.4 on LongBench v1, surpassing all R1-distilled models and QwenLong-L1-32B. At 14B, LoongRL reaches 74.2, even rivaling the much larger, heavily

Table 4: While being trained only on 16K, LoongRL generalizes impressively to context up to 128K.

| Models | NarrativeQA | | | RULER | | | |
|---|---|---|---|---|---|---|---|
| | 0-16K | 16K-32K | 32K-64K | 16K | 32K | 64K | 128K |
| Qwen2.5-7B-Instruct | 55.7 | 35.2 | 42.4 | 92.3 | 89.5 | 81.8 | 69.4 |
| R1-Distill-Qwen-7B | 55.7 | 35.2 | 42.4 | 18.9 | 4.4 | 1.4 | 0.9 |
| **LoongRL-7B** | **69.8** | **47.4** | **57.2** | **93.4** | **91.4** | **86.2** | **76.8** |
| Qwen2.5-14B-Instruct | 55.7 | 40.7 | 48.3 | 93.4 | 92.5 | 82.3 | 73.6 |
| R1-Distill-Qwen-14B | 63.0 | 35.9 | 54.6 | 85.7 | 82.0 | 60.2 | 28.2 |
| R1-Distill-Qwen-32B | 57.4 | 44.4 | 58.9 | 90.3 | 88.9 | 71.5 | 40.9 |
| QwenLong-L1-32B | 65.9 | 48.1 | 60.0 | 87.6 | 86.8 | 80.6 | 70.2 |
| **LoongRL-14B** | **69.5** | **55.2** | **64.3** | **95.4** | **95.1** | **87.1** | **79.9** |

trained o3-mini (74.5) and DeepSeek-R1 (74.9). **(ii)** Our KeyChain-driven RL proves far more effective than existing methods. It improves Qwen2.5-7B-Instruct and Qwen2.5-14B-Instruct by **+23.5%** and **+21.1%**, respectively. In contrast, R1-distilled Qwen models, trained on long-CoT reasoning data, yield a modest +11.8% gain at 14B and even degrade 7B performance by -17.7%. Similarly, QwenLong-L1-32B, trained via conventional long-context RL on R1-distill-Qwen-32B, improves by just +4.6% on average. Notably, LoongRL-7B even outperforms QwenLong-L1-32B by +2.3%, demonstrating that much smaller models can surpass larger baselines with our approach.

Table 3 compares the performance on HELMET, a challenging benchmark that includes **out-domain** long-context generation tasks such as RAG, citation-grounded generation, and long-document summarization. Our models consistently outperform all baselines, further demonstrating that the reasoning patterns induced by LoongRL generalize effectively to diverse long-context scenarios.

**Training at short, generalize better to long**. The strong results in Table 2 are largely driven by KeyChain data, which enables our models to acquire a plan-retrieve-reason-recheck thinking pattern. Although trained on 16K input contexts, this patterns generalizes effectively to much longer contexts. As shown in Table 4, both LoongRL-7B and 14B achieve substantial gains on longer-context reasoning and retrieval benchmarks. On NarrativeQA (32K-64K), they achieve impressive absolute gains of +14.8% and +16.0%, respectively, far exceeding R1-distilled models and QwenLong-L1-32B, which are trained with much longer contexts. On the RULER benchmark (up to 128K), while other baselines degrade sharply with increasing context length, our models maintain consistently strong performance, showing that the learned reasoning pattern transfers robustly to longer contexts.

**Near-lossless general short reasoning**. Table 2 also reports LoongRL's performance on short-context reasoning and general tasks, showing that it effectively preserves the base models' capabilities. On MMLU, LoongRL even yields gains of **+2.8%** (7B) and **+1.1%** (14B). In contrast, both R1-distilled models and QwenLong-L1-32B suffer performance drops. On instruction following (IFEval), R1-distilled models degrade sharply (-16.5% at 7B, -8.4% at 14B), while LoongRL shows only minimal declines (-0.3% and -2.6%). For math reasoning, although R1-distilled models benefit from heavy long-CoT data distillation, our approach stably preserves the base models' math ability.

**Improved long-context retrieval**. We evaluate the impact of different approaches on retrieval using the Needle in a Haystack benchmark, which measures a model's ability to find "needles" from long documents at varying depths. As shown in Fig. 3, the base Qwen2.5-7B-Instruct fails to fully pass this benchmark. In contrast, our LoongRL improves retrieval substantially, with LoongRL-7B achieving perfect accuracy across all depths. Other approaches remain limited, with R1-Distill-7B unable to retrieve beyond 20K and even the larger QwenLong-L1-32B failing to achieve a full pass.

## 4.3 ABLATION STUDY

**Multi-stage RL training sustains improvements**. To understand how LoongRL achieves strong performance, we report step-by-step gains and average training lengths for 7B and 14B across the three RL stages. As shown in Fig. 4 (c,d), average response length steadily increases throughout training. Fig. 4 (a,b) presents long-context reasoning accuracy, which grows consistently across each stage, demonstrating the effectiveness of the multi-stage RL curriculum.

**Ablation on the KeyChain data**. Our KeyChain training data effectively encourages models to acquire new long-context reasoning patterns during RL. To evaluate its effectiveness, we replace it with an equal amount of regular long-context multi-hop QA data on Qwen2.5-7B-Instruct while

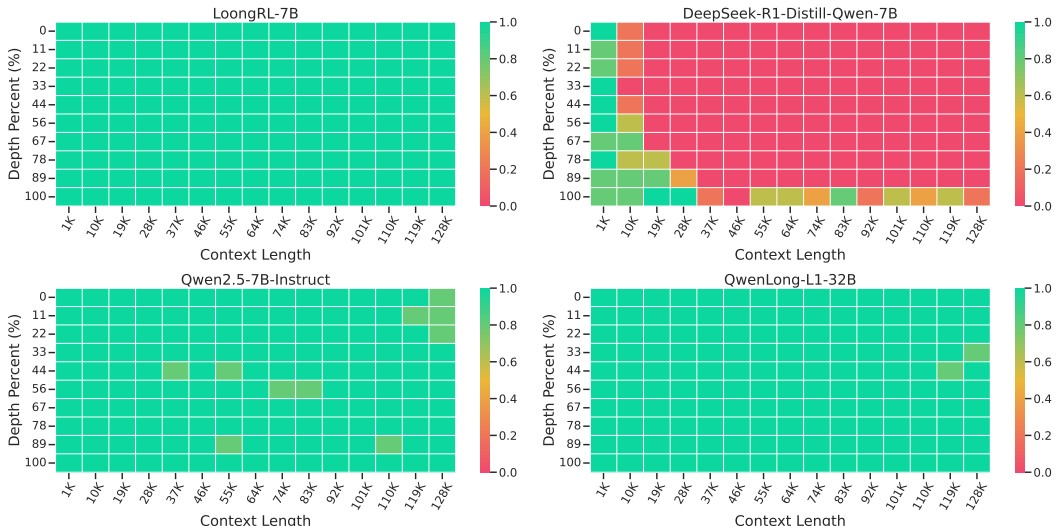

Figure 3: Needle in a Haystack retrieval across document depths. The base Qwen2.5-7B-Instruct does not fully pass the benchmark, whereas LoongRL-7B achieves perfect 100% retrieval accuracy.

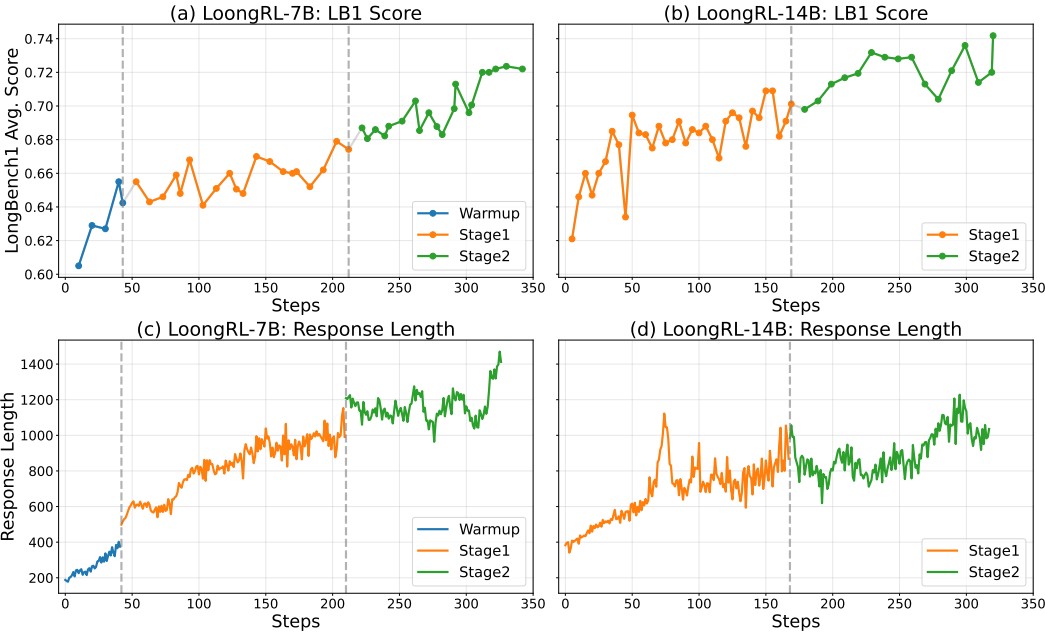

Figure 4: Long-context reasoning accuracy and training response lengths throughout RL training.

keeping all other RL settings identical. Table 5 shows the comparison results. RL with regular QA data yields moderate gains (66.2), whereas incorporating KeyChain data drives a substantial leap to 72.4, reaching frontier-level performance.

Moreover, as shown in Fig. 1(b), models trained with regular long-context multi-hop data exhibit a mixed reasoning-with-retrieval pattern. They often lack an explicit planning step and do not perform careful reason over the retrieved information, making them more prone to errors. This demonstrates that KeyChain not only significantly enhances long-context reasoning but also unlocks capabilities that cannot be achieved with conventional QA data, highlighting its unique and critical role.

**Ablation on the answer verifier**. To evaluate our two-way substring exact match for verifying answer correctness, we compare it with three widely used baselines on Qwen2.5-7B-Instruct: (i) F1 score between extracted answers and the ground truth (Shi et al., 2025; Chuang et al., 2025); (ii) LLM-as-a-judge using DeepSeek-V3 to assess consistency with the ground truth; and (iii) ex-

Table 5: Ablation study on the effectiveness of KeyChain data.

| Models | HotpotQA | 2WikiMultiHopQA | MuSiQue | NarrativeQA | QASPER | Avg. |
|---|---|---|---|---|---|---|
| Qwen2.5-7B-Instruct | 69.5 | 50.5 | 34.0 | 44.5 | 46.0 | 48.9 |
| LoongRL-7B (no KeyChain Data) | 80.3 | 84.7 | 58.5 | 53.0 | 54.5 | 66.2 |
| **LoongRL-7B** | **83.1** | **91.1** | **65.6** | **58.4** | **63.6** | **72.4** |

Table 6: Ablation study on the different answer verifiers on the 7B.

| Reward Verifier | HotpotQA | 2WikiMultiHopQA | MuSiQue | NarrativeQA | QASPER | Avg. |
|---|---|---|---|---|---|---|
| F1 score | 79.5 | 86.4 | 58.0 | 46.6 | 55.0 | 65.1 |
| LLM-as-a-judge | 80.0 | 87.6 | 60.0 | 52.3 | 54.5 | 65.2 |
| Exact match | 82.7 | **91.3** | **66.3** | 51.0 | 54.9 | 69.2 |
| **Two-way Substring Exact Match (ours)** | **83.1** | 91.1 | 65.6 | **58.4** | **63.6** | **72.4** |

act match, requiring extract answer to match the ground truth exactly. As shown in Table 6, F1 and LLM-as-a-judge yield moderate gains, while exact match performs better but is overly strict, penalizing essentially correct answers with minor formatting differences. In contrast, our two-way substring exact match maintains high precision while allowing variations, boosting long-context reasoning scores to 72.4 and clearly demonstrating its practical reliability for RL training.

## 5 CONCLUSION

This work introduces LoongRL, a data-driven reinforcement learning approach for advanced long-context reasoning. By creating a novel dataset, KeyChain, which transforms standard multi-hop questions into high-difficulty tasks, LoongRL trains models to develop a "plan-retrieve-reason-recheck" thinking pattern. A key finding is that this emergent reasoning ability generalizes remarkably well. Models trained on 16K token contexts can effectively solve tasks up to 128K tokens. Our resulting LoongRL-14B model achieves a 74.2 score on long-context QA benchmarks, rivaling much larger frontier models like o3-mini and DeepSeek-R1. These significant gains are achieved while successfully preserving the model's crucial short-context reasoning and retrieval capabilities.

## REPRODUCIBILITY STATEMENT

We have made extensive efforts to ensure the reproducibility of our work. Details of the GRPO algorithm and hyperparameters are provided in Section 3.2.1 and Section 4.1. We provide our training prompt template in Appendix A.2. The datasets used in our experiments are described in Table 1. To further facilitate reproducibility, the supplementary materials include (i) our RL training code, (ii) the code for synthesizing KeyChain data, and (iii) several representative samples of the synthesized KeyChain data. These resources, together with the descriptions in the main text and appendix, provide all necessary information for replicating our results.

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

# A APPENDIX

## A.1 USE OF LARGE LANGUAGE MODELS IN PAPER WRITING

In this work, we used large language models (LLMs) solely as general-purpose tools. Specifically, we employed LLMs to improve the clarity and readability of the paper. Additionally, during our ablation experiments, we evaluated the effectiveness of the answer verifier by using DeepSeek-V3 as the baseline in an LLM-as-a-judge setting.

## A.2 TRAINING PROMPT TEMPLATE

For reproducibility, we include the exact prompt format used during training (see Figure 5). The model was trained to first generate intermediate reasoning enclosed in `<think> ... </think>`, and then provide the final answer enclosed in `\boxed{}`.

---

**System Prompt**

A conversation between User and Assistant. The User asks a question, and the Assistant solves it. The Assistant first thinks about the reasoning process in the mind and then provides the User with the answer. The reasoning process is enclosed within `<think> </think>` and answer is enclosed within `\boxed{}` tags, respectively, i.e., `<think>` reasoning process here `</think>` `\boxed{answer here}`.

---

Figure 5: System prompt used during training

## A.3 EXAMPLE OF KEYCHAIN-AUGMENTED TRAINING DATA

Figure 6 shows a sample skeleton of our training data, illustrating how we carry out the Key-Chain augmentation for long-context reinforcement learning.

---

Please read the following text.
Document 0:
…
Document 3:
Who's Who? is a studio album by American jazz musician John Scofield. It features two different bands, one acoustic and one electric. The acoustic group, featuring Scofield's then-employer Dave Liebman on saxophones, Eddie G\u00f3mez on bass, and Billy Hart on drums, recorded "The Beatles" and "How the West Was Won". …
{"bdd640fb-0667-4ad1-9c80-317fa3b1799d": "23b8c1e9-3924-46de-beb1-3b9046685257"}.
…
Document 10:
…
The university is one of the smallest of the 23 CSU campuses in California. Sonoma State offers 92 Bachelor's degrees, 19 Master's degrees, one Doctoral degree (Doctor of Education), and 11 teaching credentials. {"972a8469-1641-4f82-8b9d-2434e465e150": "Musician and satirist Allie Goertz wrote a song about the "The Simpsons" character Milhouse, who Matt Groening named after who?"}.
…
Document 47:
Neil Affleck
{"bd9c66b3-ad3c-4d6d-9a3d-1fa7bc8960a9": "972a8469-1641-4f82-8b9d-2434e465e150"}.
Neil Affleck (born 1953) is a Canadian animator, director, and former actor. He has worked as an animator on "The Simpsons" and "Family Guy", and as an actor appeared in a leading role in the 1981 film "My Bloody Valentine". {"9a1de644-815e-46d1-bb8f-aa1837f8a88b": "b74d0fb1-32e7-4629-8fad-c1a606cb0fb3"}.
…
In the context above, there is one correct question to answer. The correct question can only be found by following the correct consecutive chain of key:value pairs encoded with UUID strings (e.g., f81d4fae-7dec-11d0-a765-00a0c91e6bf6), starting from "bdd640fb-0667-4ad1-9c80-317fa3b1799d".
Find the correct question first, then answer it.

---

Figure 6: A skeleton of KeyChain-augmented training data used in LoongRL.

### A.4 COMPARISON BETWEEN TRACES OF KEYCHAIN VS. NON-KEYCHAIN MODELS

Here we contrast two models trained with same hyperparameters but different data. The KeyChain-augmented model shows structured, step-by-step reasoning (colored as green) and clear re-checking when ambiguity arises, while the non-KeyChain model often shortcuts retrieval and proceeds to reasoning prematurely, leading to errors (colored as red). The following traces illustrate these differences on the same question:

**Question:** Gary L. Bennett was a part of the space missions that have a primary destination of what celestial body?

---

**KeyChain-trained Model**

To determine the primary destination of the space missions that Gary L. Bennett was a part of, we need to identify the space missions he worked on from the given passages and then find the primary destination of those missions.

From Passage 1, we can see that Gary L. Bennett worked on the following space missions: "His professional career has included work on the Voyager, Galileo, and Ulysses space missions,"

Now, we need to find the primary destination of these missions. Let's check Passage 5 for information about the Ulysses mission:

"Ulysses ( yoo-LISS-eez, UK also YOO-liss-eez) was a robotic space probe whose primary mission was to orbit the Sun and study it at all latitudes."

Therefore, the primary destination of the Ulysses space mission is the Sun.

Although we should check the other missions mentioned (Voyager and Galileo) for completeness, we can see from Passage 3:

"No missions were planned to coincide with the transit of Earth visible from Mars on 11 May 1984 and the Viking missions had been terminated a year previously. Consequently, the next opportunity to observe such an alignment will be in 2084." "On 21 December 2012, the Cassini-Huygens probe, in orbit around Saturn, observed the planet Venus transiting the Sun." "These are Schiaparelli EDM lander, the Mars Polar Lander, and Deep Space 2. These were all sterilized for surface missions but not for special regions (Viking pre-sterilization only). Mars Polar Lander, and Deep Space 2 crashed into the polar regions which are now treated as special regions because of the possibility of forming liquid brines."

And from Passage 4: "Ulysses (spacecraft)" "The spacecraft was originally named Odysseus, because of its lengthy and indirect trajectory to study the solar poles. It was renamed Ulysses, the Latin translation of "Odysseus", at ESA's request in honor not only of Homer's mythological hero but also of Dante's character in the Inferno. Ulysses was originally scheduled for launch in May 1986 aboard the Space Shuttle Challenger on STS-61-F. Due to the 28 January 1986 loss of Challenger, the launch of Ulysses was delayed until 6 October 1990 aboard Discovery (mission STS-41)."

The Voyager mission is known to have a primary destination of the outer planets, specifically Jupiter, Saturn, Uranus, and Neptune. The Galileo mission had a primary destination of Jupiter.

However, the question specifically asks about the space missions that Gary L. Bennett was a part of, and from the information provided, the Ulysses mission is clearly stated to have the Sun as its primary destination.

Therefore, the primary destination of the space missions that Gary L. Bennett was a part of, specifically the Ulysses mission, is the Sun.

the Sun

---

**No KeyChain Model**

(Lack of systematic decomposition of the question)

First, we need to identify the space missions mentioned in the passage that Gary L. Bennett was involved with. From the biography provided, we can see that Bennett was involved in several space missions, including the Voyager, Galileo, and Ulysses space missions. The question asks for the primary destination of these missions.

- For the Voyager mission, the passage states: "He was the flight safety manager for the radioisotope power sources currently in use on the Voyager 1 and Voyager 2 spacecraft (which went to Jupiter, Saturn, Uranus, Neptune and beyond)."

- For the Galileo mission, it mentions: "Prior to coming to NASA, Bennett held key positions in DoE's space radioisotope power program, including serving as Director of Safety and Nuclear Operations for the radioisotope power sources that were used on the Galileo mission to Jupiter and that are being used on the Ulysses mission to explore the polar regions of the Sun."

- For the Ulysses mission, the passage states: "Ulysses ( yoo-LISS-eez, UK also YOO-liss-eez) was a robotic space probe whose primary mission was to orbit the Sun and study it at all latitudes."

From these statements, we can deduce that Bennett was involved in missions to Jupiter, Saturn, Uranus, Neptune, and the Sun. However, the primary destinations mentioned directly in relation to Bennett's roles are Jupiter for the Galileo and Ulysses missions, and multiple planets including Jupiter, Saturn, Uranus, and Neptune for the Voyager missions.

Given the specific mention of Bennett's role in the Galileo mission to Jupiter, and the broader statement about the Ulysses mission studying the Sun, Jupiter stands out as a primary destination that Bennett was directly involved with.

Jupiter

## A.5 TRAJECTORY EXAMPLES ON LONG-CONTEXT RETRIEVAL

We show trajectories to highlight the impact of KeyChain data on general long-context retrieval. In the Ruler VT examples in A.5, the KeyChain-trained model demonstrates clearer, stepwise retrieval, while the non-KeyChain-trained model conflates intermediate variable assignments and propagates errors, showing KeyChain data's role in eliciting structured and correct retrieval behaviors.

---

**RL Trajectory Example with KeyChain Data: Long-Context Retrieval on Ruler VT**

<think> However, the specific value 92018 is only directly assigned to one variable in the text provided. Let's break it down step by step:
1. The first assignment in the text is: "VAR QPE = 64886 The grass is green. The sky is blue. The sun is yellow. Here we go. There and back again." This assigns the value 64886 to the variable QPE.
2. The next assignments form a chain: "VAR SEJ = VAR QPE The grass is green. The sky is blue. The sun is yellow. Here we go. There and back again." This assigns the value of QPE (which is 64886) to the variable SEJ.
3. "VAR ZQO = VAR SEJ The grass is green. The sky is blue. The sun is yellow. Here we go. There and back again." This assigns the value of SEJ (which is 64886) to the variable ZQO.
4. "VAR RVU = VAR ZQO The grass is green. The sky is blue. The sun is yellow. Here we go. There and back again." This assigns the value of ZQO (which is 64886) to the variable RVU.
5. "VAR FAI = VAR RVU The grass is green. The sky is blue. The sun is yellow. Here we go. There and back again." This assigns the value of RVU (which is 64886) to the variable FAI.
6. The value 92018 is assigned to the variable SGMLJ: "VAR SGMLJ = 92018 The grass is green. The sky is blue. The sun is yellow. Here we go. There and back again."
7. Then, it forms a new chain: "VAR PBDME = VAR SGMLJ The grass is green. The sky is blue. The sun is yellow. Here we go. There and back again." This assigns the value of SGMLJ (which is 92018) to the variable PBDME.
8. "VAR EANSM = VAR PBDME The grass is green. The sky is blue. The sun is yellow. Here we go. There and back again." This assigns the value of PBDME (which is 92018) to the variable EANSM.
9. "VAR QPKBX = VAR EANSM The grass is green. The sky is blue. The sun is yellow. Here we go. There and back again." This assigns the value of EANSM (which is 92018) to the variable QPKBX.
10. "VAR YYZJM = VAR EANSM The grass is green. The sky is blue. The sun is yellow. Here we go. There and back again." This assigns the value of EANSM (which is 92018) to the variable YYZJM.
So, the variables that are assigned the value 92018 are: SGMLJ, PBDME, EANSM, QPKBX, and YYZJM. </think>
<think> The variables that are assigned the value 92018 are:
1. VAR SGMLJ = 92018 2. VAR PBDME = VAR SGMLJ 3. VAR EANSM = VAR PBDME 4. VAR QP-KBX = VAR EANSM5. VAR YYZJM = VAR EANSM Thus, the variables are SGMLJ, PBDME, EANSM, QPKBX, and YYZJM. </think>
\boxed{SGMLJ, PBDME, EANSM, QPKBX, YYZJM}

---

**RL Trajectory Example WITHOUT KeyChain Data: Long-Context Retrieval on Ruler VT**

<think> The reasoning process involves identifying the lines where the value 92018 is first assigned to a variable, which is "VAR PBDME = 92018". Then, tracking this value through all subsequent assignments where this variable is reassigned to other variables. The variables that inherit this value through the assignments are EANSM, YYZJM, SGMLJ, and FAI. Since the question asks for variables assigned the value 92018, and FAI is assigned 92018 through the chain VAR PBDME = VAR SGMLJ, VAR SGMLJ = 92018, and so on, the final list of variables is PBDME, EANSM, YYZJM, SGMLJ, and FAI. </think>
\boxed{PBDME, EANSM, YYZJM, SGMLJ, FAI}

---

## A.6 LOONGRL TRAINING DYNAMICS

Fig. 7 and Fig. 8 show the training dynamics for the 7B and 14B models, respectively. Unlike the 7B setting, the 14B model is trained using only two curriculum stages, *distractor* and *hard-mined*, skipping the initial warm-up due to its strong base capabilities. As shown in two figures, a similar cyclical trend emerges: counts of consistently correct predictions and average rewards rise within each stage, reset when transitioning to a harder task pool, and increase again. This multi-stage training continues to provide informative learning signals and prevents saturation, while the steady growth in response length indicates that the model persistently extends its retrieval and reasoning chains in long-context multi-hop reasoning.

## A.7 LONGBENCH-V2

We evaluate the LoongRL series and baseline models on the LongBench-v2 benchmark. For models without long chain-of-thought (Long CoT) reasoning capability, we follow the original LongBench-

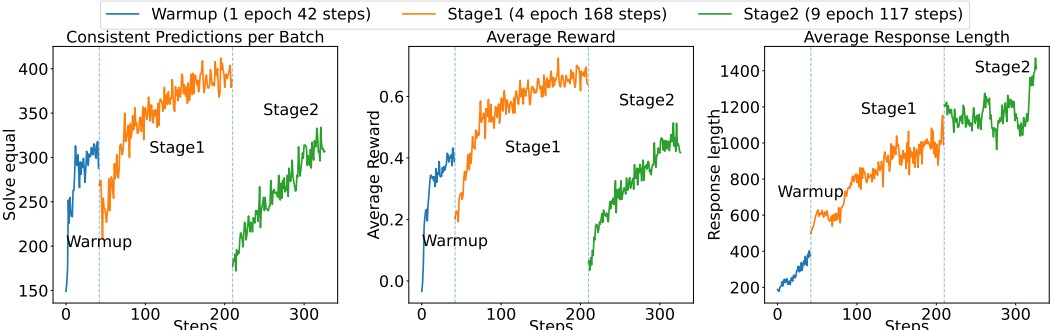

Figure 7: Training metrics for the three-stage schedule. Vertical dashed lines mark the transitions *Warmup → Stage II (distractor-augmented)* and *Stage II → Stage III (hard-mined)*; stage lengths correspond to our setup (about 42, 168, and 117 steps).

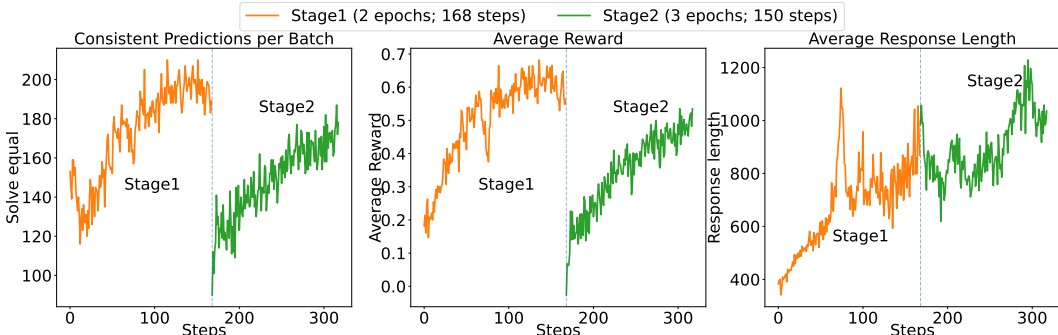

Figure 8: Training metrics for the LoongRL-14B's two-stage schedule. Vertical dashed lines mark the transitions *Stage I (distractor-augmented)* and *→ Stage III (hard-mined)*; stage lengths correspond to our setup (about 168, and 150 steps).

Table 7: Comparison of LoongRL models with other baselines on the LongBench-v2 benchmark, grouped by Difficulty, Length, and Task Type.

| Model | Overall | Difficulty | | Length | | | Task Type | | | | | |
| --- | --- | --- | --- | --- | --- | --- | --- | --- | --- | --- | --- | --- |
| | | Easy | Hard | Short | Medium | Long | Long ICL | Long SDU | Code | SingleDoc QA | Long Dialogue | MultiDoc QA |
| o3-mini | 46.4 | 52.9 | 42.4 | **56.1** | 41.2 | 40.2 | 43.2 | 40.6 | 46.0 | 46.8 | **71.8** | 41.6 |
| GPT-4o | 48.3 | **61.8** | 40.3 | 46.2 | 48.4 | **51.0** | **57.6** | 44.4 | **66.7** | 46.2 | 50.0 | 40.8 |
| QwQ-32B | **51.2** | 57.8 | **47.1** | **53.7** | **51.2** | 46.5 | 54.6 | 35.6 | 50.4 | **51.8** | 56.9 | **49.9** |
| R1-Distill-LLaMa-70B | 34.2 | 35.4 | 33.4 | 47.2 | 28.4 | 24.1 | 28.4 | 15.2 | 32.0 | 37.7 | 46.2 | 35.2 |
| Qwen2.5-7B-Instruct | 31.2 | 32.3 | 30.5 | **42.8** | 24.7 | 25.0 | 25.9 | 30.3 | 42.0 | 35.4 | 35.9 | 23.2 |
| R1-Distill-Qwen-7B | 27.0 | 29.2 | 25.7 | 30.6 | 23.7 | 27.8 | 21.0 | 18.2 | 32.0 | 25.1 | 33.3 | **32.0** |
| **LoongRL-7B** | **36.2** | **41.1** | **33.1** | 40.6 | **34.4** | **32.4** | **35.8** | **39.4** | **44.0** | **38.9** | **59.0** | 21.6 |
| Qwen2.5-14B-Instruct | 35.3 | 34.9 | 35.5 | 43.3 | 32.6 | 27.1 | 33.8 | 33.3 | 32.0 | 38.3 | 35.9 | 33.6 |
| R1-Distill-Qwen-14B | 36.2 | 40.2 | 33.8 | 44.1 | 31.4 | 32.6 | 36.8 | 36.4 | 28.4 | 38.4 | 44.1 | 33.4 |
| R1-Distill-Qwen-32B | 38.6 | 40.1 | 37.6 | 48.9 | 33.5 | 31.5 | 29.6 | 39.4 | 38.0 | 39.4 | 51.3 | 39.2 |
| QwenLong-L1-32B | 40.8 | **46.4** | 37.4 | **52.4** | 35.7 | 31.5 | 37.0 | 32.7 | **43.2** | 40.0 | 55.4 | **41.0** |
| **LoongRL-14B** | **42.3** | **46.4** | **39.9** | 44.4 | **43.3** | **37.0** | **39.5** | **45.5** | 38.0 | **44.0** | **59.0** | 37.6 |

v2 CoT setting, using a temperature of $0.1$, sampling five responses per query, and reporting the average score across them (*Avg@5*). For models with Long CoT reasoning ability, we instead adopt a temperature of $0.6$, again sampling five responses and reporting *Avg@5*. In addition, for the Qwen family and our LoongRL models, we apply the YaRN method to extend the context length to 128k tokens. The overall comparison results are summarized in Table 7.

## A.8 RULER

We evaluated the retrieval capabilities of models on long-text tasks using the RULER benchmark. For models without long-context reasoning abilities, we followed the original RULER setting by appending a prompt suffix designed to guide the model to produce completion-style answers, e.g. "What is the special magic number for wandering-age mentioned in the provided text? The special magic number for wandering-age mentioned in the provided text is". In contrast, for models capable of long-context reasoning, we removed this completion-style suffix, as preliminary experiments

Table 8: RULER benchmark results across different context lengths. For QwQ, QwenLong, Qwen2.5 model series, we report their YaRN variants for 64k and 128k.

| Model | 4k | 8k | 16k | 32k | 64k | 128k | Avg. |
|---|---|---|---|---|---|---|---|
| o3-mini | 96.58 | 96.85 | 94.69 | 90.85 | 74.81 | 65.40 | 86.53 |
| DeepSeek-R1 | 98.46 | 97.98 | 97.18 | 96.06 | 94.92 | 85.10 | 94.95 |
| GPT-4o | 97.69 | 96.73 | 96.73 | 96.02 | 94.46 | 89.10 | 95.12 |
| QwQ-32B (YaRN@64k/128k) | 89.10 | 86.46 | 83.84 | 78.42 | 64.72 | 59.68 | 77.37 |
| R1-Distill-LLaMa-70B | 94.89 | 95.60 | 93.75 | 89.60 | 79.65 | 0.00 | 75.58 |
| Llama3.1-70B-Instruct | 96.78 | 96.64 | 95.82 | 94.87 | 89.21 | 64.53 | 89.64 |
| R1-Distill-LLaMa-8B | 83.89 | 79.80 | 73.77 | 64.46 | 51.06 | 1.28 | 59.04 |
| Llama3.1-8B-Instruct | 96.10 | 93.81 | 90.91 | 86.73 | 84.77 | 74.15 | 87.75 |
| Qwen2.5-7B-Instruct (YaRN@64k/128k) | **95.16** | 93.73 | 92.31 | 89.46 | 81.79 | 69.41 | 86.31 |
| R1-Distill-Qwen-7B | 65.70 | 48.29 | 18.86 | 4.38 | 1.41 | 0.88 | 23.25 |
| LoongRL-7B (YaRN@64k/128k) | 95.06 | **94.34** | **93.37** | **91.36** | **86.18** | **76.84** | **89.53** |
| Qwen2.5-14B-Instruct (YaRN@64k/128k) | 96.27 | 95.11 | 93.38 | 92.53 | 82.33 | 73.57 | 88.86 |
| R1-Distill-Qwen-14B | 91.44 | 86.29 | 85.73 | 82.00 | 60.24 | 28.23 | 72.32 |
| R1-Distill-Qwen-32B | 93.61 | 91.64 | 90.27 | 88.90 | 71.51 | 40.88 | 79.47 |
| QwenLong-L1-32B (YaRN@64k/128k) | 91.71 | 88.51 | 87.55 | 86.81 | 80.64 | 70.19 | 84.24 |
| LoongRL-14B (YaRN@64k/128k) | **97.56** | **96.14** | **95.36** | **95.11** | **87.14** | **79.92** | **91.87** |

indicated that these models tend not to provide direct completions but instead perform explicit reasoning before answering. After removing the suffix, we allowed the reasoning-capable models to generate up to 8192 tokens and subsequently extracted the model's answer from the text following the "`</think>`" token in its output.

## A.9 NEEDLE-IN-A-HAYSTACK

We further evaluated the *needle-in-a-haystack* (NIAH) task, which specifically measures the retrieval ability of models in extremely long-text settings. Figure 9 reports the performance of our LoongRL-14B model. Results demonstrate that LoongRL-14B maintains strong retrieval accuracy across extended context lengths, showcasing its robustness for long-context information retrieval.

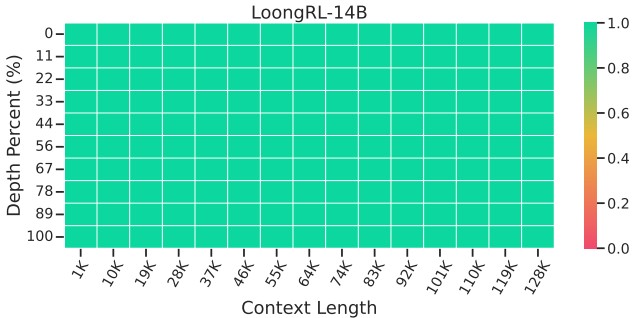

Figure 9: Needle-in-a-Haystack performance of LoongRL-14B.

## A.10 UUID ABLATION

Table 9: Analysis of the design choice of KeyChain data. The RL experiments are conducted on Qwen2.5-7B-Instruct.

| KeyChain data | Avg. | HotpotQA | 2WikiMultiHopQA | MuSiQue | NarrativeQA | QASPER |
|---|---|---|---|---|---|---|
| UUID (Reported) | **72.4** | 83.1 | 91.1 | 65.6 | 58.4 | 63.6 |
| Random String | 72.2 | 81.5 | 92.3 | 65.5 | 58.5 | 63.3 |

As described in Section 3.1, we use UUIDs as unique identifiers for variables in the KeyChain data. To ensure that the observed performance gains are not tied to the specific UUID format, we conduct an ablation study in which UUIDs are replaced with random strings of the same length. As shown in Table 9, this modification results in nearly identical performance, indicating that the keys do not

need to follow a specific UUID format; they only need to be high-entropy and non-semantic to prevent shortcut learning.

## A.11 EMERGING SKILLS ANALYSIS DURING RL

Table 10: Example responses of LoongRL-14B at different RL training steps, showing how retrieval, reasoning, rechecking, and planning behaviors evolve over training.

| Stage | Ckpt at different training steps | Response Length | Skills | Accuracy |
|---|---|---|---|---|
| Stage 1 | Step 10 | 2,228 | Retrieve + Reason | × |
| | Step 20 | 2,963 | Retrieve + Reason + **Recheck** | ✓ |
| | Step 70 | 5,685 | Retrieve + Reason + Recheck | ✓ |
| | Step 118 | 6,101 | Retrieve + Reason + Recheck | ✓ |
| Stage 2 | Step 199 | 1,480 | **Plan** + Retrieve + Reason | ✓ |
| | Step 229 | 2,596 | **Plan** + Retrieve + Reason + **Recheck** | ✓ |
| | Step 319 | 4,143 | **Plan** + Retrieve + Reason + Recheck | ✓ |

To understand the content of model responses and how LoongRL progressively acquires reasoning skills, we manually inspected LoongRL-14B outputs across RL training steps and observed a clear evolution in reasoning behavior.

As shown in Table 10, early in training (Step 10), the model attempts retrieval and reasoning but often fails. Over time, it develops a retrieve–reason–recheck pattern, producing longer and more accurate responses (e.g., 2K → 6K tokens). By Step 118, responses are even longer due to repeated retrieval, reasoning, and rechecking, though the model still struggles to answer correctly. After difficulty-focused Stage 2 training (Step 199), explicit planning emerges before retrieval and reasoning, yielding shorter responses that correctly solve the tasks. Continued training consolidates these abilities, producing the full plan–retrieve–reason–recheck pattern that integrates structured planning, accurate retrieval, reasoning, and self-verification.

In summary, longer responses primarily reflect the emergence of rechecking and iterative reasoning, while Stage 2 introduces explicit planning, completing the full reasoning pattern.

