# OpenReview forum: "LoongRL: Reinforcement Learning for Advanced Reasoning over Long Contexts"
_ICLR.cc/2026/Conference — ICLR 2026 Oral_

### Official Review · Reviewer_ZCVK · 2025-10-30

**Soundness:** 4
**Presentation:** 4
**Contribution:** 4
**Rating:** 8
**Confidence:** 4

**Summary:**

This paper introduces LoongRL, a data-driven reinforcement learning (RL) method to improve long-context reasoning in LLMs, tackling the dual challenges of scarce high-difficulty data and high computational costs. The core contribution is KeyChain, a novel data synthesis technique that transforms short multi-hop QA into difficult long-context tasks by hiding the true question behind a multi-step UUID chain within a distractor-filled document. The authors find that training with RL on KeyChain data induces an emergent "plan-retrieve-reason-recheck" pattern that, critically, generalizes from 16K training contexts to 128K test contexts. This method yields substantial accuracy gains on Qwen2.5 models (+21-23%), allowing them to rival much larger frontier models while preserving short-context reasoning and retrieval abilities.

**Strengths:**

1. The KeyChain synthetic data is a clever and novel approach to enable long-context RL training. This work shows adding tracing and distractors in the multi-hop QA data can improve the reasoning of long-context tasks.
2. The authors show models trained on 16K contexts can generalize to 128K, directly addressing the prohibitive computational cost of RL rollouts at full context length.
3.  The authors designed their training data mix to prevent catastrophic forgetting of short-context and retrieval skills. The results in Table 2 (showing stable MMLU, MATH, and IFEval scores) and Figure 3 (perfect NIAH scores) are crucial for demonstrating that this specialization does not come at an unacceptable cost.

**Weaknesses:**

1. The testing benchmark (LongBench and RULER) and training dataset are almost in the same domain (e.g., Wikipedia). We can see large improvement on these two benchmarks but relatively small improvement on LongBenchv2. It would be better to show more long-context benchmark improvements like HELMET, AA-LCR, MRCR, etc.
2. The underlying reasoning data is derived entirely from multi-hop QA datasets (HotpotQA, MuSiQue, etc.). It is unclear if the emergent reasoning pattern would generalize to other long-context tasks, such as summarizing a novel or analyzing a large codebase.

**Questions:**

1. Since you have included RULER QA, multi-key, multi-value NIAH in the training set, then can we still evaluate on RULER in Table 3? I think model may already learn the test data distribution in your RL training.
2. Can we try training on HotPotQA keychain data but evaluate on other dataset like MuSiQue to see whether the training data can help to generalize in unseen dataset? Also, I am interested if your data can generalize your model to unseen domain more than Wikipedia.
3. In Figure 4, you showed model learned to increase response length. Can you explain what includes in the response? Do model first learn to plan, then retrieve, reason and recheck? What skills do model learn progressively during training?

---

> ### Author Response · Authors · 2025-11-21
> **Rebuttal by Authors**
>
> Thank you very much for your thoughtful and highly positive review, as well as the strong scores across soundness, presentation, and contribution. We sincerely appreciate your encouraging assessment of LoongRL.  We have carefully considered all concerns raised and provide our clarifications and responses below.
>
> >### The testing benchmark (LongBench and RULER) and training dataset are almost in the same domain (e.g., Wikipedia). We can see large improvement on these two benchmarks but relatively small improvement on LongBenchv2. It would be better to show more long-context benchmark improvements like HELMET, AA-LCR, MRCR, etc.  Can we try training on HotPotQA keychain data but evaluate on other dataset like MuSiQue to see whether the training data can help to generalize in unseen dataset? Also, I am interested if your data can generalize your model to unseen domain more than Wikipedia.
>
> **Response**: Thank you for your thoughtful suggestions. We understand the importance of clarifying whether the evaluation tasks are in- or out-of-distribution relative to our training data. We address your concerns from three perspectives and will clarify these points more explicitly in the revision.
>
> 1) *Out-of-distribution evaluation tasks in LongBench v1 and RULER*. As described in paper, our training data is mainly derived from Wikipedia multi-hop QA. However, several evaluation tasks in LongBench and RULER are actually out-of-domain. For instance, Qasper consists of scientific papers, and NarrativeQA is based on novels and film scripts. RULER also includes categories such as multi-hop tracing (VT) and aggregation that do not appear in our RL training data, and the evaluation is conducted at 128K context length while our RL data are limited to 16K. LoongRL still achieves substantial gains on these tasks, indicating that the learned reasoning pattern generalizes beyond the Wikipedia domain and beyond the training context length.
>
> 2) *Improvements on LongBenchv2 are meaningful given the benchmark's difficulty*. LongBench v2 contains many tasks that are not represented in our training data and are highly challenging. These tasks span long in-context learning, long dialogue history understanding, reasoning over code repositories, and interpreting long-structured data such as table QA and knowledge graph reasoning. While the absolute improvement on LongBench v2 may appear smaller compared to LongBench v1 and RULER, it is important to note that frontier-level reasoning models also achieve relatively low scores on these tasks due to the high difficulty. Under this challenging setting, LoongRL-14B performs on par with models such as Qwen3-30B-A3B-Thinking and Kimi-K2-Instruct, demonstrating substantial relative gains and meaningful generalization to harder and more diverse long-context tasks.
>
> |Model|LongBench v2|
> | :--:   |:--:   |
> |o3-mini| 46.4|
> |Kimi-K2-Instruct|44.3|
> |Qwen3-30B-A3B-Thinking| 42.5|
> |**LoongRL-14B**|42.3 (+7)|
> |QwenLong-L1-32B| 40.8|
> |o1-mini|38.9|
> |**LoongRL-7B**|36.2 (+5)|
>
> 3) *Significant improvements on additional long-context benchmark HELMET*. Following your suggestion, we additionally evaluated LoongRL on the HELMET benchmark, covering **RAG**, **generations with citations**, and **summarization**.  The results are shown below:
>
>
> |Model| Avg.| RAG (Natural Questions)| RAG (HotpotQA)| RAG (PopQA)| RAG (TriviaQA)| Generation with Citations (ALCE ASQA)| Generation with Citations (ALCE Qampari)| Summarization (InfiniteBench Sum)|Summarization (Multi-LexSum)|
> | :--:   |:--:   |:--:   |:--:   |:--:   |:--:   |:--:   |:--:   |:--:   |:--:   |
> |Qwen2.5-7B-Instruct| 22.82| 21.20| 20.30|27.70|53.30|6.07|0.07|14.70|39.24|
> |DeepSeek-R1-Distill-Qwen-7B| 6.73| 4.67| 4.00| 19.17| 9.17| 11.62| 5.25| 0.00| 0.00|
> |**LoongRL-7B**|**44.78 (+21.96)** | **50.00**|**60.00**| **53.67**| **82.80**| **15.01**| **7.81**| **28.09**| **60.83**|
> |Qwen2.5-14B-Instruct| 40.55|40.50|48.00|45.30|81.70|24.09|6.63|27.71|50.44|
> |DeepSeek-R1-Distill-Qwen-14B|29.68|41.80|41.70|40.80|32.00|14.86|9.75|16.08|40.45|
> |QwenLong-L1-32B| 41.79|46.50|44.66|35.33|85.33|16.33|12.17|28.19|**65.77**|
> |**LoongRL-14B**|**48.96 (+8.41)**| **55.33**|**60.33**|**51.00**|**88.17**|**22.98**|**14.35**|**33.73**| 65.76|
>
> These results demonstrate that LoongRL achieves substantial improvements across a diverse set of more challenging and out-domain long-context tasks such as RAG, summarization and citation-grounded generation. This provides additional evidence that the reasoning patterns incentivized through LoongRL generalizes effectively to broader long-context scenarios.
>
> We hope these clarifications address your concern, and we will incorporate them into the revised version for improved clarity.
>
> [1] HELMET: How to evaluate long-context language models effectively and thoroughly. https://arxiv.org/abs/2410.02694

---

> ### Author Response · Authors · 2025-11-21
> **Rebuttal by Authors**
>
> >### Since you have included RULER QA, multi-key, multi-value NIAH in the training set, then can we still evaluate on RULER in Table 3? I think model may already learn the test data distribution in your RL training.
>
> **Response**:  Thank you for raising this important concern. We would like to clarify the motivation behind our design of the RULER-related training data and explain why evaluating on RULER remain valid.
>
> Initially, we synthesized multi-key and multi-value retrieval-based training data with the goal of using RL to enhance the model’s ability to retrieve relevant information from long contexts and to verify whether training on shorter contexts could generalize to longer ones. While these synthetic data may resemble RULER-style tasks, we carefully designed them to avoid any test-set leakage with the RULER benchmark. Specifically, for the filler text, we used passages from PG19, which has a distribution distinct from RULER. For task selection, we considered only the two retrieval-focused types, multi-key-value and key-multi-value, whereas RULER contains 13 subtasks covering broader reasoning categories such as multi-hop tracing and aggregation. The training input length was up to 16K tokens, while the RULER evaluation uses up to 128K context.
>
> Despite these differences, LoongRL demonstrates substantial improvement across RULER subtasks, including those not directly represented in the training data. We will include these clarifications in the revision to ensure that the design and generalization results are clearly communicated.
>
>
> >### In Figure 4, you showed model learned to increase response length. Can you explain what includes in the response? Do model first learn to plan, then retrieve, reason and recheck? What skills do model learn progressively during training?
>
> **Response**: We appreciate your insightful question. To understand what the response contains and how LoongRL progressively acquires reasoning skills, we manually inspected LoongRL-14B responses across RL training steps and observed a clear progression in reasoning behavior, summarized below:
>
> | Checkpoint at different training steps | Length | Skills | Accuracy |
> |------------|--------|--------|----------|
> | Stage1 Step 10 | 2,228 | Retrieve + Reason | ✗ (0.00) |
> | Step 20 | 2,963 | Retrieve + Reason + **Recheck** | ✓ (1.00) |
> | Step 70 | 5,685 | Retrieve + Reason + Recheck | ✓ (1.00) |
> | Step 118 | 6,101 | Retrieve + Reason + Recheck | ✓ (1.00) |
> | **Stage 2 (difficulty-focused training) Step 199** | 1,480 | **Plan** + Retrieve + Reason | ✓ (1.00) |
> | Step 229 | 2,596 | **Plan** + Retrieve + Reason + **Recheck** | ✓ (1.00) |
> | Step 319 | 4,143 | **Plan** + Retrieve + Reason + Recheck | ✓ (1.00) |
>
> Early in training (i.e., step 10), the model attempts retrieval and reasoning but often fails. Over time, it acquires the retrieve–reason–recheck pattern, leading to higher accuracy and longer responses (e.g., 2K → 6K characters). At Step 118, responses are much longer due to repeated retrieval, reasoning, and rechecking while still struggling to answer correctly. After difficulty-focused Stage 2 training (Step 199), the model introduces explicit planning before retrieval and reasoning, producing shorter responses that correctly solve the tasks. Further training consolidates these skills, yielding the full plan–retrieve–reason–recheck pattern that integrates structured planning, accurate retrieval, reasoning, and self-verification.
>
> In summary, longer responses primarily reflect the emergence of rechecking and iterative reasoning, and Stage 2 introduces explicit planning that completes the full reasoning pattern.
>
> We appreciate your question as it helped us better understand the model’s skill progression throughout RL training.

---

> > ### Comment · Reviewer_ZCVK · 2025-11-26
> >
> > Thank you very much for your responses. Additional HELMET results and emerging skills analysis during RL have strengthen the contribution. I will keep my current rating score and tend to accept the paper.

---

> > > ### Author Response · Authors · 2025-11-27
> > > **Thank you for your follow-up!**
> > >
> > > Thank you for your follow-up. We are glad that the HELMET results and the RL emergent-skills analysis strengthened the contribution. We appreciate your constructive feedback and will incorporate these improvements into the revision.

---

### Official Review · Reviewer_kHZc · 2025-10-30

**Soundness:** 4
**Presentation:** 3
**Contribution:** 4
**Rating:** 8
**Confidence:** 2

**Summary:**

- Problem: Current RL-for-reasoning mostly targets short contexts; long-context scenarios require both retrieval from large inputs and multi-step reasoning. Direct long-context RL rollouts are costly; high-quality, verifiable, and challenging long-context data are scarce.
- Key idea: A data-driven RL pipeline (LoongRL) centered on KeyChain, a synthesis method that transforms standard multi-hop QA into high-difficulty long-context tasks by inserting UUID key→value chains that hide the true question among distractors. Solving requires: follow chain → recover the real question → retrieve relevant evidence → reason → answer.
- RL method: Train with GRPO on ~16K-token inputs to induce advanced patterns that purportedly generalize to 128K without full-length RL. Reward is outcome-only, via a rule-based “two-way substring exact match” applied to a boxed final answer to reduce reward hacking and tolerate formatting variants. A balanced data mix and multi-stage schedule preserve short-context capabilities.
- Emergent pattern: Empirically, RL over KeyChain elicits a plan–retrieve–reason–recheck behavior that improves reliability and long-context generalization beyond training lengths.
- Results: On Qwen2.5-7B/14B, LoongRL delivers large gains on long-context multi-hop QA (e.g., +23.5% and +21.1% absolute on LongBench v1), matches or approaches larger frontier models (o3-mini, DeepSeek-R1) at much smaller scales, improves Needle-in-a-Haystack and RULER (up to 128K), and preserves/mostly maintains general short-context abilities (MMLU, MATH, IFEval).
- Ablations: (i) Replacing KeyChain with regular long-context QA yields notably lower gains; (ii) two-way substring match outperforms F1, LLM-as-judge, and strict EM as a training reward; (iii) multi-stage curriculum helps.

**Strengths:**

- Clear, focused objective and thoughtful problem decomposition:
  - Tackles a real gap: moving beyond retrieval to robust long-context reasoning.
  - Designs data and training to be verifiable and compute-aware (shorter rollouts, longer generalization).
- Novel and pragmatic data construction:
  - KeyChain is a neat way to force chain tracing and disambiguation under heavy distractors, requiring both retrieval and reasoning.
  - Uses real QA seeds (HotpotQA, MuSiQue, 2Wiki) to ground tasks in natural language rather than entirely synthetic QA.
- RL methodology matched to constraints:
  - Outcome-only rule-based reward (boxed answer + two-way substring) is a strong, reproducible alternative to LLM-as-judge, with an ablation showing its advantage for training.
  - GRPO and small KL with a staged curriculum is standard yet well-justified to stabilize training.
- Strong empirical results at favorable scale/cost:
  - Large absolute gains vs. strong baselines on LongBench v1, competitive with much larger models.
  - Convincing length generalization from 16K training to 128K evaluation across NarrativeQA, RULER, and Needle-in-a-Haystack.
  - Maintains short-context and instruction-following capabilities, often outperforming R1-distilled counterparts that degrade on these.
- Diagnostic evidence:
  - Ablations disentangle the contribution of KeyChain vs. conventional long-context QA.
  - Qualitative trajectories demonstrate the claimed plan–retrieve–reason–recheck behavior.
- Reproducibility:
  - Clear training details, datasets, prompts, and claim of releasing code and sample data.

**Weaknesses:**

- Synthetic structure risk:
  - The KeyChain format (UUID chains with a designated starting key) has a highly regular, explicit structure. There is a risk the model learns to exploit these patterns rather than developing general long-context reasoning skills. Although downstream improvements suggest transfer, additional tests against structurally varied chains would better establish robustness.
- Reward and evaluation concerns:
  - Two-way substring match is pragmatic but may still admit false positives (e.g., partial overlap on ambiguous entities) or false negatives (paraphrases). The ablation is helpful, but more analysis on reward precision/recall would increase confidence against reward hacking.
  - Inference differences across baselines (temperature, sampling, pass@1 vs pass@k) and the use of YaRN to extend contexts for only some models can complicate fairness. While partially addressed, I recommend additional matched settings or a dedicated “strictly matched” comparison table.
- Overlap and contamination:
  - Training uses multi-hop QA seeds overlapping with evaluation domains (e.g., HotpotQA). The paper does not explicitly state safeguards for train/test separation or whether exact question/passages overlap with benchmark test items after augmentation. Without a rigorous de-duplication policy, there is a risk of leakage.
- Limited statistical characterization:
  - No variance across seeds or CI reported; improvements are large but stability under re-runs is not quantified.
- Claims of emergent patterns:
  - The plan–retrieve–reason–recheck behavior is shown qualitatively. A quantitative metric (e.g., automated tagging of plan/recheck segments; retrieval step correctness rates) would make the claim stronger and trackable across training stages and datasets.
- Scope and breadth:
  - Most evaluations are reading comprehension/multi-hop QA and retrieval. It would be informative to include tasks where the query is obscured in different ways (tables, code bases, non-UUID chains), or tool-augmented settings (e.g., retrieval tools outside the context), to demonstrate broader generalization.

**Questions:**

- Data and leakage
  - Please detail the exact splitting and de-duplication strategy to ensure that augmented KeyChain instances do not overlap (content-wise) with evaluation items in LongBench v1/v2. How do you prevent direct reuse of the same question/evidence pairs?
  - When filling long contexts with distractors (including reusing documents from filtered tasks), how do you ensure no evaluation leakage?
- Reward design
  - Provide analyses of reward false positives/negatives: e.g., on a held-out set with human-verified correctness, report precision/recall for two-way substring vs EM/F1/LLM-as-judge. Include examples of where two-way substring fails and mitigation (e.g., normalization, entity linking).
  - Clarify the reference policy used for KL (π_ref): is it the base instruct model frozen? Any mixing with SFT references?
- Fairness and settings
  - Add a “strictly matched inference” comparison where all models use the same temperature, pass@1, and no YaRN (or apply YaRN uniformly where possible) on a subset of tasks, to isolate the contribution of LoongRL vs. differences in decoding/context-extensions.
  - Report sensitivity to decoding hyperparameters (temperature/top-p) on the main long-context benchmarks.

---

> ### Author Response · Authors · 2025-11-21
> **Rebuttal by Authors**
>
> We sincerely thank the reviewer for the thorough and positive evaluation of our work. We greatly appreciate the recognition of our KeyChain design, RL methodology, and empirical results on long- and short-context benchmarks. Your questions and suggestions have provided valuable insights and will significantly improve the clarity and quality of our paper. We have carefully considered all questions raised and provide our clarifications and responses below.
>
> > ### Data and leakage
> Please detail the exact splitting and de-duplication strategy to ensure that augmented KeyChain instances do not overlap (content-wise) with evaluation items in LongBench v1/v2. How do you prevent direct reuse of the same question/evidence pairs?
> When filling long contexts with distractors (including reusing documents from filtered tasks), how do you ensure no evaluation leakage?
>
> **Response**: Thank you for raising this important question! We take data decontamination very seriously and have implemented strict procedures to ensure that neither the KeyChain training instances or their distractor contexts overlap with our evaluation benchmarks.
>
> First, for HotpotQA, 2WikiMultiHopQA, and MuSiQue, all our data are constructued exclusively from **the official training splits** of the respective datasets. No test examples are used. Distractor documents are also sampled only from our filtered training-split documents.
>
> Second, to minimize any possible overlap with LongBench v1 and v2, we apply two de-duplication steps. First, we remove any exact string matches between our synthesized training queries (including questions and retrieved evidence) and evaluation benchmarks. Second, we apply 8-gram overlap detection and filter out any samples to eliminate potential partial overlaps. This procedure is applied both to the augmented key–value instances and to the distractor documents used when filling long contexts.
> We will clarify these details in the revised version. We hope these clarifications address your concerns regarding data contamination.
>
> >### Fairness and settings
> Add a “strictly matched inference” comparison where all models use the same temperature, pass@1, and no YaRN (or apply YaRN uniformly where possible) on a subset of tasks, to isolate the contribution of LoongRL vs. differences in decoding/context-extensions. Report sensitivity to decoding hyperparameters (temperature/top-p) on the main long-context benchmarks.
>
> **Response**: Thank you for the helpful suggestion. We fully agree that fair evaluation requires careful control of inference configurations. We have already taken steps to ensure fairness and consistency across models, following the widely adopted principle that *reasoning models and non-reasoning instruct models should be evaluated using their respective best-practice inference settings*.
>
> Concretely, all reasoning models, including DeepSeek-R1, and our LoongRL models, share the same inference configuration: temperature 0.6, top-p 0.95, up to 128K input tokens and 10K output tokens, with eight samples for average pass@1. These settings are standard for reasoning-style decoding across recent reasoning models.  For QwenLong-L1, we follow the settings reported in its paper (i.e., temperature 0.7, top-p 0.95, max 10K response length). Non-reasoning instruct models such as Qwen2.5-Instruct use temperature 0, consistent with their official practice. For tasks requiring more than 32K context,  YaRN is applied uniformly to all models. We will add a more detailed description of these settings in the revision for clarify and transparency.
>
> >### Limited statistical characterization:
> No variance across seeds or CI reported; improvements are large but stability under re-runs is not quantified.
>
> **Response**: Thank you for the valuable suggestion. For RL training, due to the high GPU cost, we conducted all experiments using a fixed training seed (42). For evaluation, our original setup also used a fixed seed (42). However, since reasoning models are inherently stochastic, we adopted a more robust evaluation setting: temperature = 0.6, top-p = 0.95, generating 8 samples per instance and reporting the average pass@1.
>
> Following your suggestion, we further evaluated LoongRL-7B under four additional evaluation seeds. The results are shown below:
>
> |Run|Seed|Avg. | Hotpotqa|2WikiMultiHopQA|MuSiQue|NarrativeQA|Qasper|
> | :--: | :--:   |:--:   |:--:   |:--:   |:--:   |:--:   |:--:   |
> |Run-1 (Reported)|42 |72.4| 83.1|91.1|65.6|58.4|63.6|
> |Run-2| 30867|72.0|82.5|92.8|63.9|57.7|63.3|
> |Run-3|10687 |72.2| 82.8|92.3|64.9|57.8|63.1|
> |Run-4|7686 |71.3| 81.5|90.5|62.5|58.5|63.5|
> |Run-5|13591 |72.4| 85.0|90.1|67.0|56.9|63.2|
>
> These results indicate that the variance across evaluation seeds is minimal, showing that LoongRL's improvements are stable and reproducible. We will include these results in the revision for completeness and clarity.

---

> ### Author Response · Authors · 2025-11-21
> **Rebuttal by Authors**
>
> >## Reward design
> Provide analyses of reward false positives/negatives: e.g., on a held-out set with human-verified correctness, report precision/recall for two-way substring vs EM/F1/LLM-as-judge. Include examples of where two-way substring fails and mitigation (e.g., normalization, entity linking).
> Clarify the reference policy used for KL (π_ref): is it the base instruct model frozen? Any mixing with SFT references?
>
> **Response**: Thank you for your insightful questions! We answer your questions as follows:
>
> 1) **Analysis of reward design**: Before presenting the results, we would like to clarify the motivation behind our two-way exact match. This reward is specifically designed for RL training. Compared with LLM-as-a-judge or F1 score, our reward is stricter, effectively preventing false positives where an incorrect model output is mistakenly accepted. This helps reduce the risk of reward hacking during training. Compared with strict exact match, it reduces false negatives. While some false negatives may still occur, we consider **avoiding false positives more critical in the RL training**. This design choice is further supported by our ablation results in Table 5.
>
>
> And, following your suggestion, we conducted a human-verified evaluation on 50 randomly sampled trajectories. Each answer was manually labeled as "correct/incorrect", and we compared four reward signals: F1 (>0.5), Exact Match, LLM-as-a-judge, and our two-way substring exact match. The results are summarized below:
>
> |Method| TP| TN| FP| FN|Accuracy|Precision|Recall|
> | :--:   |:--:   |:--:   |:--:   |:--:   |:--:   |:--:   |:--:   |
> |F1 Score (>0.5)| 26|15|0|9|82%|100%|74%|
> |Exact Match| 27|15|0|8|84%|100%|77%|
> |LLM-as-a-judge| 34|12|3|1|92%|91.9%|**97.1%**|
> |**Substring Exact Match (Ours)**|32|15|0|3|**94%**|**100%**|91%|
>
> As shown, our two-way match achieves **zero false positives**, which is essential for preventing reward hacking during RL, while maintaining high recall (91%), substantially higher than Exact Match (77%) and F1 (74%). Although LLM-as-a-judge achieves slightly higher recall, it introduces false positives, making it unsuitable as a reward signal. Overall, our method attains the highest accuracy (94%), providing a strict yet robust reward signal for RL training. These results will be added to the revised version.
>
>
> **Examples**: We show three examples:
>
> Row 1: The model’s answer is correct;  only LLM-as-a-judge identifies it. Our method produces a false negative.
>
> Row 2: The model’s answer is incorrect; LLM-as-a-judge generates a false positive, but our method correctly rejects it.
>
> Row 3: The model’s answer is correct; exact match fails due to phrasing variation; our two-way method correctly identifies correctness.
>
> |Question|Ground-truth| Model answer|Manual Label| F1| Exact match| LLM as a judge| Our two-way substring exact match|
> | :--: | :--:   |:--:   |:--:   |:--:   |:--:   |:--:   |:--:   |
> |Where does the witch live?|The Atlas Mountains|on atlas' mountain, within a cavern, by a secret fountain|✓|0|0|1|0 (false negative)|
> |Celebrity Fifteen to One has had more than one appearance by an English writer and former Conservative Member of what?|Parliament|house of commons|✗ |1|1|0 (false positive)|1|
> |The Huskies football team were invited to the Alamo Bowl where they were defeated by a team coached by Art Briles and who played their home games at what statium?|Floyd Casey Stadium|floyd casey stadium in waco, texas|✓|1|0 (false negative)|1|1|
>
>
> 2) **KL reference policy**: our reference model is the base instruct model (Qwen2.5-7B-instruct and Qwen2.5-14B-instruct) frozen, without any additional SFT. We will clarify this in the revision.
>
> >### Claims of emergent patterns:
> The plan–retrieve–reason–recheck behavior is shown qualitatively. A quantitative metric (e.g., automated tagging of plan/recheck segments; retrieval step correctness rates) would make the claim stronger and trackable across training stages and datasets.
>
> **Response**: We appreciate your insightful suggestion. We agree that a quantitative metric would strength our claim and allow systematic tracking across training stages and datasets. We attempted to perform automated evaluation using an LLM, but found the results to be unreliable. We plan to explore more robust and reliable methods for quantifying this emergent behavior in future work.

---

> ### Author Response · Authors · 2025-11-21
> **Rebuttal by Authors**
>
> >### Scope and breadth:
> Most evaluations are reading comprehension/multi-hop QA and retrieval. It would be informative to include tasks where the query is obscured in different ways (tables, code bases, non-UUID chains), or tool-augmented settings (e.g., retrieval tools outside the context), to demonstrate broader generalization.
>
> **Response**: We appreciate your constructive suggestion! In our original paper, we evaluated LoongRL on LongBench v2, which already includes several non-multi-hop QA tasks such as code repository understanding and Long SDU (structured-data reasoning over long JSON-like inputs), as shown in Appendix Table 6. LoongRL achieves significant improvements on these tasks, demonstrating its ability to generalize beyond standard multi-hop QA and retrieval.
>
> To further test broader generalization, we additionally evaluated LoongRL on HELMET [1], a more challenging long-context generation benchmark. HELMET includes demanding non-QA tasks such as RAG, generation with citations, and long-document summarization. Our results show that LoongRL consistently improves performance across these tasks, providing strong evidence that the reasoning patterns learned through RL generalize beyond the original multi-hop QA-focused setting.
>
> |Model| Avg.| RAG (Natural Questions)| RAG (HotpotQA)| RAG (PopQA)| RAG (TriviaQA)| Generation with Citations (ALCE ASQA)| Generation with Citations (ALCE Qampari)| Summarization (InfiniteBench Sum)|Summarization (Multi-LexSum)|
> | :--:   |:--:   |:--:   |:--:   |:--:   |:--:   |:--:   |:--:   |:--:   |:--:   |
> |Qwen2.5-7B-Instruct| 22.82| 21.20| 20.30|27.70|53.30|6.07|0.07|14.70|39.24|
> |DeepSeek-R1-Distill-Qwen-7B| 6.73| 4.67| 4.00| 19.17| 9.17| 11.62| 5.25| 0.00| 0.00|
> |**LoongRL-7B**|**44.78 (+21.96)** | **50.00**|**60.00**| **53.67**| **82.80**| **15.01**| **7.81**| **28.09**| **60.83**|
> |Qwen2.5-14B-Instruct| 40.55|40.50|48.00|45.30|81.70|24.09|6.63|27.71|50.44|
> |DeepSeek-R1-Distill-Qwen-14B|29.68|41.80|41.70|40.80|32.00|14.86|9.75|16.08|40.45|
> |QwenLong-L1-32B| 41.79|46.50|44.66|35.33|85.33|16.33|12.17|28.19|**65.77**|
> |**LoongRL-14B**|**48.96 (+8.41)**| **55.33**|**60.33**|**51.00**|**88.17**|**22.98**|**14.35**|**33.73**| 65.76|
>
>
> [1] HELMET: How to evaluate long-context language models effectively and thoroughly. https://arxiv.org/abs/2410.02694

---

### Official Review · Reviewer_WxW6 · 2025-10-31

**Soundness:** 3
**Presentation:** 3
**Contribution:** 2
**Rating:** 4
**Confidence:** 3

**Summary:**

The paper's core contribution is an RL training task designed to elicit reasoning behaviours that are adapted to long-context problems. The authors augment existing multi-hop QA tasks by adding additional documents to the problem context and constructing key-value chains that need to be traversed for the appropriate question to be retrieved and then answered based on the documents' content. Experiments show that this task composition incentivises models to learn a structured plan, retrieve, reason, re-check reasoning pattern for long-context tasks. Two-way substring matching is used for verifiable rewards. A mixture of KeyChain, standard QA, retrieval and math reasoning tasks are used for training to improve long-context reasoning and maintain short-context reasoning performance. Results show that short-context reasoning is indeed preserved and 16k training contexts lead to long-context reasoning abilities that generalise to 128k evaluation contexts. Needle-in-a-Haystack evaluations are passed with 100% accuracy, demonstrating strong retrieval performance.

**Strengths:**

The paper addresses an important open problem: reasoning over long contexts beyond basic retrieval. By enabling RL to target nontrivial but verifiable long-context reasoning problems, the KeyChain dataset overcomes a key bottleneck of long-context RL finetuning. The results indicate strong empirical performance gains on long-context benchmarks without regressing on the short-context reasoning benchmarks considered, however the latter is expected given the inclusion of short-context reasoning tasks in the training mixture.

**Weaknesses:**

The KeyChain data construction is highly task-specific: synthetic multi-hop QA with UUID breadcrumbs. It is unclear from the current results if the learned reasoning behaviour generalises to other domains, such as open-ended dialogue, summarisation, or multi-document synthesis. There is no evaluation on long-context generation tasks, which has been artificially decoupled from reasoning over long, static input contexts. A major claim is the emergence of a general reasoning pattern for long-context problems, but it is unclear whether the structured reasoning pattern that is qualitatively demonstrated arises from task scaffolding (i.e., explicit chain following) rather than self-organised reasoning behaviour. Finally, the ablations are currently incomplete, particularly with regards to the multi-stage curriculum. Whilst accuracy is showing to rise across stages, there is no direct comparison between the full curriculum and: skipping the warm-up, doing a single RL stage on all data, or removing the Stage II difficulty filtering. The wam-up stage is even omitted for the 14B model, but the impact of this on results is never shown.

**Questions:**

How did you detect or measure the plan, retrieve, reason, recheck reasoning pattern?

Could you show how much each curriculum stage contributes independently?

Have you tested LoongRL on non-QA long-context tasks (e.g., summarisation, code reasoning, document comparison)?

How do the training costs for LoongRL compare to training QwenLong-L1?

---

> ### Author Response · Authors · 2025-11-21
> **Rebuttal by Authors**
>
> Thank you for your thoughtful review and the time you dedicated to our work! We appreciate your recognition of our KeyChain design, RL methodology, and strong empirical results. Below, we provide clarifications and additional experiments that address the concerns you raised.
>
> > ### The KeyChain data construction is highly task-specific: synthetic multi-hop QA with UUID breadcrumbs. It is unclear from the current results if the learned reasoning behaviour generalises to other domains, such as open-ended dialogue, summarisation, or multi-document synthesis. Have you tested LoongRL on non-QA long-context tasks (e.g., summarisation, code reasoning, document comparison)?
>
> **Response**: Thank you for your questions. We would like to clarify that **although our current experiments primarily focus on QA tasks, the KeyChain mechanism is neither task specific nor QA specific, and the incentivized long-context reasoning patterns generalize well across diverse and challenging tasks**. We address this from four perspectives:
>
> 1) *LoongRL is not QA-specific; KeyChain is a general RL training mechanism for long-context reasoning*. Its goal is to provide verifiable rewards for RL to incentivize a more advanced long-context reasoning pattern, i.e., the resulting plan-retrieve-reason-recheck pattern. These capabilities are fundamental to a broad range of long-context query-response task (e.g., fact retrieval, document reasoning, etc). QA is simply a convenient training domain due to the availability of suitable seed data; the underlying mechanism is task-agnostic.
>
> 2) *LoongRL is not UUID-specfic; any high-entropy, non-semantic key works*. As clarified to Reviewer 6KJK, the use of UUID-like keys is not an inherent requirement. The only constraint is that keys be high-entropy and non-semantic, preventing the model from learning shortcuts based on lexical priors and ensuring explicit key–value retrieval. Our experiments with random strings yield nearly identical results. Thus, KeyChain is not restricted to the synthetic QA domain nor dependent on the specific UUID format.
>
> 3) *Empirically, LoongRL already generalizes beyond QA in both short- and long-context regimes*. For short context, Table 2 (original paper) includes evaluations on non-QA tasks such as MMLU, MATH and IFEval. For long-context, LongBench v2 (Appendix Table 6) contains tasks such as Code Understanding (repository-level reasoning) and Long SDU (structured-data reasoning over long JSON-like inputs). These tasks require skills that are fundamentally different from knowledge-based QA; the multiple-choice format is used only to support reliable automatic evaluation.
>
> 4) **Additional evaluation on HELMET, a challenging long-context generation benchmark**. To further address your concern, we evaluate LoongRL on HELMET[1], which includes demanding non-QA tasks such RAG, generation with citations, and long-document summarization. The results are shown below:
>
> |Model| Avg.| RAG (Natural Questions)| RAG (HotpotQA)| RAG (PopQA)| RAG (TriviaQA)| Generation with Citations (ALCE ASQA)| Generation with Citations (ALCE Qampari)| Summarization (InfiniteBench Sum)|Summarization (Multi-LexSum)|
> | :--:   |:--:   |:--:   |:--:   |:--:   |:--:   |:--:   |:--:   |:--:   |:--:   |
> |Qwen2.5-7B-Instruct| 22.82| 21.20| 20.30|27.70|53.30|6.07|0.07|14.70|39.24|
> |DeepSeek-R1-Distill-Qwen-7B| 6.73| 4.67| 4.00| 19.17| 9.17| 11.62| 5.25| 0.00| 0.00|
> |**LoongRL-7B**|**44.78 (+21.96)** | **50.00**|**60.00**| **53.67**| **82.80**| **15.01**| **7.81**| **28.09**| **60.83**|
> |Qwen2.5-14B-Instruct| 40.55|40.50|48.00|45.30|81.70|24.09|6.63|27.71|50.44|
> |DeepSeek-R1-Distill-Qwen-14B|29.68|41.80|41.70|40.80|32.00|14.86|9.75|16.08|40.45|
> |QwenLong-L1-32B| 41.79|46.50|44.66|35.33|85.33|16.33|12.17|28.19|**65.77**|
> |**LoongRL-14B**|**48.96 (+8.41)**| **55.33**|**60.33**|**51.00**|**88.17**|**22.98**|**14.35**|**33.73**| 65.76|
>
> These results demonstrate that LoongRL achieves substantial improvements across a diverse set of long-context tasks beyond multi-hop QA, including summarization and citation-grounded generation. This provides additional evidence that the reasoning patterns acquired through LoongRL transfer effectively to broader long-context scenarios.
>
> We hope these clarifications address your concern, and we will incorporate them into the revised version for improved clarity.
>
> [1] HELMET: How to evaluate long-context language models effectively and thoroughly. https://arxiv.org/abs/2410.02694

---

> ### Author Response · Authors · 2025-11-21
> **Rebuttal by Authors**
>
> >### A major claim is the emergence of a general reasoning pattern for long-context problems, but it is unclear whether the structured reasoning pattern that is qualitatively demonstrated arises from task scaffolding (i.e., explicit chain following) rather than self-organised reasoning behaviour. How did you detect or measure the plan, retrieve, reason, recheck reasoning pattern?
>
> **Response**: Thank you for your thoughtful question. Below, we address your concern from four aspects:
>
> 1) *No scaffolding in the training data*. LoongRL is trained on Qwen2.5-7B-Instruct and Qwen2.5-14B-Instruct without any additional SFT. The RL data include only a long context, a query, and a final answer. We do not provide any step-by-step demonstrations, templates, or structured reasoning examples. Because the reward depends solely on the final answer, the model cannot learn the plan-retrieve-reason-recheck pattern by imitating any explicit structure in the data.
>
> 2) *No scaffolding in prompts*. The RL prompt template (Appendix Figure 5) only specifies the $<$think$>$ and $<$answer$>$ sections. We do not use instructions such as “first plan, then retrieve,” numbered steps, or any staged reasoning templates. All evaluations strictly use the standard prompts of each benchmark, with no modifications. Therefore, the emergent structured reasoning is not tied to prompt engineering.
>
> 3) *Strong performance improvements across diverse tasks*. If the pattern were induced by scaffolding, it would only manifest in tasks that match that scaffold format. Instead, LoongRL exhibits broad generalization. Specifically, LoongRL achieves strong improvements on LongBench v1, LongBench v2, RULER and the long-context generation benchmark HELMET. Moreover, the 7B LoongRL model even approaches or matches much larger frontier models.  This cross-task emergence indicates a big shift in internal reasoning behavior rather than template imitation.
>
> 4) *How we detect the reasoning pattern*. We first observed qualitative changes after RL: large performance gains paired with noticeably more structured traces. We attempted automated detection with an LLM but found it unreliable due to both false positives and false negatives, so we performed human inspection. We randomly sampled 20 traces from LoongRL-14B and compared them to QwenLong-L1-32B. Among these,  QwenLong-L1-32B generally shows no explicit planning, and retrieval and reasoning are mixed, as illustrated in paper Figure 1. LoongRL-14B (18 out of 20 traces) consistently begins with an explicit plan, follows it with organized retrieval, performs structured reasoning, and self-initiates re-checks on harder problems. Since neither the training data nor the prompts contain such structure, these behaviors reflect self-organized, RL-induced long-context reasoning rather than scaffolded imitation. Finally, we note that developing quantitative metrics to track this pattern systematically is an important direction and leave this as future work.
>
> **We have also included the 20 sampled traces from both LoongRL-14B and QwenLong-L1-32B in the supplementary zip file for reference**. We hope these clarifications address your concern, and we will incorporate them into the revision for improved clarity.

---

> ### Author Response · Authors · 2025-11-21
> **Rebuttal by Authors**
>
> >### Finally, the ablations are currently incomplete, particularly with regards to the multi-stage curriculum. Whilst accuracy is showing to rise across stages, there is no direct comparison between the full curriculum and: skipping the warm-up, doing a single RL stage on all data, or removing the Stage II difficulty filtering. The warm-up stage is even omitted for the 14B model, but the impact of this on results is never shown. Could you show how much each curriculum stage contributes independently?
>
> **Response**: Thank you for highlighting the importance of ablations on the curriculum design. We fully agree that understanding the contribution of each stage is valuable. Due to limited GPU resources, our original experiments prioritized evaluating the core contribution of LoongRL, i.e., the KeyChain-augmented long-context reasoning RL data.  While curriculum training is also beneficial, its effectiveness has been widely demonstrated in prior work [1,2].
>
> To directly address your concern, we have now conducted additional ablation experiments on the 7B model. We removed (1) the warmup stage and (2) Stage II difficulty filtering, while keeping the total training steps the same as in our main experiment. The results, shown in the table below, indicate that both the warmup stage and Stage II are important for improving long-context reasoning performance. These additional results will be included in the revised version to provide a more complete picture of curriculum contributions.
>
> |RL stages| Avg. | HotpotQA|2WikiMultiHopQA| MuSiQue| NarrativeQA| QASPER|
> | :--:   |:--:   |:--:   |:--:   |:--:   |:--:   |:--:   |
> |Stage1+Stage2 (no warmup stage)| 68.7 |82.0| **92.4**| 64.6| 49.8|54.6|
> |Warmup Stage+Stage1 (no stage2)| 68.8| 81.6| 89.2| 62.6| 56.1| 54.4|
> |Warmup Stage+Stage1+Stage2 (Ours)| **72.4**|**83.1**|91.1|**65.6**|**58.4**|**63.6**|
>
> [1] rStar2-Agent: Agentic Reasoning Technical Report. https://arxiv.org/abs/2508.20722
>
> [2] The art of scaling reinforcement learning compute for LLMs. https://arxiv.org/pdf/2510.13786
>
> >### How do the training costs for LoongRL compare to training QwenLong-L1?
>
> **Response**: Thank you for your question! Although QwenLong-L1 does not report exact GPU hours, we can provide a rough comparison based on the information available in their paper:
>
>
> |Model|SFT before RL? |Input length| Maximum output length| GPUs| GPU hours|
> | :--:   |:--:   |:--:   |:--:   |:--:   |:--:   |
> |QwenLong-L1| Yes| 20K in phase I, 60K in phase  II| 10K | 32xA100 GPUs| Not reported|
> |LoongRL-7B| No | 16K| 4K| 16xA100 GPUs| 150 h |
> |LoongRL-14B| No| 16K |4K|8xMI300X GPU | 140 h |
>
> Overall, LoongRL requires fewer GPUs (8-16 vs 32), significantly shorter input lengths (16K vs. 60K), and shorter maximum output lengths during RL rollouts (4K vs. 10K). Despite these significantly lower computational requirements, LoongRL achieves strong long-context reasoning performance, demonstrating that our approach is considerably more compute-efficient compared to training QwenLong-L1. We hope this comparison clarifies the compute footprint of LoongRL relative to QwenLong-L1.
>
>
> Thank you again for your valuable feedback and suggestions. We hope these responses address your concerns and clarify any confusion, and we kindly ask you to consider re-evaluating our work.

---

> > ### Comment · Reviewer_WxW6 · 2025-11-21
> > **Responding to Rebuttal**
> >
> > Thank you for your responses, which have addressed a number of my concerns. In particular, the HELMET evaluation strengthens the paper's contribution. My reference to scaffolding was in terms of how the train task is in constructed, rather than with respect to any explicitly enforced output structure, though I appreciate the clarifications. I have raised my score to a 6.

---

> > > ### Author Response · Authors · 2025-11-23
> > > **Thank you for your follow-up!**
> > >
> > > Thank you very much for your follow-up and for updating your score. We are glad that the additional clarifications and the new HELMET evaluation helped address your concerns. We also appreciate your point regarding scaffolding in task construction; our empirical results suggest that the current design is highly effective, and we leave a deeper discussion of alternative constructions to future work.

---

### Official Review · Reviewer_6KJK · 2025-11-01

**Soundness:** 4
**Presentation:** 3
**Contribution:** 3
**Rating:** 6
**Confidence:** 3

**Summary:**

This work proposes a synthetic training data construction method to improve long context reasoning for large language models. Basic idea is to insert irrelevant documents in addition to the relevant one as a challenging dataset together with key-value chains or arbitrary key-strings, e.g., UUIDs, so that the model has to traverse key-value pairs until reaching the correct question. Experiments are carried out on standard long-context reasoning benchmarks, e.g., HotpotQA, and general benchmarks, e.g., MMLU, showing gains when compared with other large language models with long reasoning capacity.

**Strengths:**

- This work proposes an interesting approach of synthetic data construction for training a language model for long-context reasoning. Basic idea is to insert distractors both for contexts and questions so that a model has to pay attention not only the correct context, but correct question at the same time. It is interesting that a model trained only on "shorter" context, i.e., 16K, can scale to 128K contexts.
- The idea of inserting key-value pairs is quite interesting so that a model has to traverse key and value pairs until reaching the correct question. This additional complexity might enforce a model to pay more attention to the question itself during the GRPO training.
- Experiments are carried out systematically by comparing with diverse models and settings.

**Weaknesses:**

- The motivation is not clear why UUIDs are used as keys. There exist alternatives, e.g., entity names, or other random strings, could be possible.
- The detail settings are missing, e.g., the number of distractor contexts and questions, inserted to construct the synthetic dataset. It is also not clear whether the distractor questions are related to the irrelevant contexts already inserted in the long context filling step.

**Questions:**

- The performance is evaluated on the standard dataset without distractor contexts or questions, but I'm curious of the performance when such distractors are inserted for LoongRL and other models.

---

> ### Author Response · Authors · 2025-11-21
> **Rebuttal by Authors**
>
> Thank you for your thoughtful and positive feedback on our work! We sincerely appreciate your insights and your recognition of our contributions. Below, we address your specific comments.
>
> >### The motivation is not clear why UUIDs are used as keys. There exist alternatives, e.g., entity names, or other random strings, could be possible.
>
> **Response**: Thank you for the insightful question. Our motivation for using UUID-like keys is not tied to their specific format, but to ensure that the model cannot rely on any semantic prior or internal knowledge to directly predict the correct key. High-entropy, non-semantic keys prevent the model from jumping directly to the relevant segment based on lexical or semantic cues. This encourages the model to perform the explicit key-value retrieval.
>
> Also, the specific UUID format is not essential, as the keys only need to be high-entropy and non-semantic. Following your suggestion, we replaced UUIDs with uniformly random strings and conducted the full RL experiment on Qwen2.5-7B-instruct. We observed similar evaluation performance, as shown in the table below, which confirms that the more advanced long-context reasoning mechanism learned by LoongRL does not depend on the UUID format.
>
> |KeyChain|Avg. | Hotpotqa|2WikiMultiHopQA|MuSiQue|NarrativeQA|Qasper|
> | :--:   |:--:   |:--:   |:--:   |:--:   |:--:   |:--:   |
> |UUID (Reported) |**72.4**| 83.1|91.1|65.6|58.4|63.6|
> |**Random string**| **72.2**|81.5|92.3|65.5|58.5|63.3|
>
> UUIDs were chosen primarily for reproducibility and engineering convenience. We will clarify this motivation in the revised version.
>
> >### The detail settings are missing, e.g., the number of distractor contexts and questions, inserted to construct the synthetic dataset. It is also not clear whether the distractor questions are related to the irrelevant contexts already inserted in the long context filling step.
>
> **Response**: Thank you for the helpful suggestion. For each query, we construct the distractor context by randomly sampling documents of varying lengths from the filtered set (see Section 3.1, Seed Dataset Curation and Context Extension, no overlap with our training set) and concatenating them until reaching a total length of 16K tokens. Within this context, we insert 16 key–value chains pointing to 16 questions, where 15 are distractor questions and 1 is the real question. The distractor contexts and distractor questions are sampled independently for implementation convenience.
>
> We appreciate your comment and will include a full description of these settings in the revised version to improve clarity and reproducibility.
>
>
> >###  The performance is evaluated on the standard dataset without distractor contexts or questions, but I'm curious of the performance when such distractors are inserted for LoongRL and other models.
>
> **Response**: Thank you for the very interesting question. Our original main evaluations follow the standard benchmarks, which do not contain distractor contexts or questions. To directly assess the impact of distractor injection, we conducted an additional experiment by inserting key–value chains into three evaluation tasks: one in-distribution task (HotpotQA) and two out-of-distribution tasks (NarrativeQA and Qasper).
>
>
> |Model| Avg.  | HotpotQA keychain (in-distribution)| NarrativeQA keychain (out-distribution)| Qasper keychain (out-distribution)|
> | :--:   |:--:   |:--:   |:--:   |:--:   |
> |Qwen2.5-7B-Instruct|4.66 | 5.0 | 0| 9.0|
> |DeepSeek-R1-Distil-Qwen-7B| 7.2| 12.0| 2.8| 6.8|
> |**LoongRL-7B**| **55.3**| **76.6**| **34.8**| **54.4**|
> |Qwen2.5-14B-Instruct|10.3 |15.5| 2.5| 13.0|
> |DeepSeek-R1-Distil-Qwen-14B|30.3 | 39.5| 13.5| 38.0|
> |QwenLong-L1-32B|46.9 | 67.2| 27.5| 46.0|
> |**LoongRL-14B**| **61.7**| **78.9**| **47.6**| **58.7**|
>
> As shown in the table, LoongRL remains highly robust under distractor injection.  For example, LoongRL-7B achieves 76.6 on HotpotQA, 34.8 on NarrativeQA, and 54.4 on Qasper, whereas the corresponding 7B baselines (Qwen2.5-Instruct and DeepSeek-R1-Distil-Qwen-7B) collapse to single-digit performance in many cases.  A similar trend holds at the 14B scale, where LoongRL-14B significantly outperforms Qwen2.5-14B, R1-Distil-14B, and QwenLong-L1-32B.
>
> We appreciate your suggestion and will include this evaluation and its full details in the revised version.

---

> > ### Comment · Reviewer_6KJK · 2025-11-27
> >
> > Thank you for the additional explanation and experimental results. I've also read other reviews, and I'm rather leaning toward acceptance.

---

> > > ### Author Response · Authors · 2025-11-27
> > > **Thank you！**
> > >
> > > Thank you for your thoughtful follow-up and for raising the score. We truly appreciate your time and constructive feedback!

---

### Author Response · Authors · 2025-12-03
**Summary of Rebuttal Discussion**

Dear ACs and Reviewers,

Thank you very much for your valuable contributions to our work. We appreciate the reviewers’ thoughtful feedback and are encouraged by the positive overall assessment, including the recognition of the paper’s novelty and strong contributions.
In response to the insightful comments, we have conducted substantial new experiments and analyses to further strengthen the paper and improve clarity. To assist the AC and help streamline their review process, we provide a brief summary of the reviewer discussions below:

|Reviewer| Initial score| Updated score| Notes (Quotes from discussions)|
| :--:   |:--:   |:--:   |:--:   |
|Reviewer 6KJK|6|8|"I've also read other reviews, and I'm rather leaning toward acceptance"|
|Reviewer WxW6|4|6|"I have raised my score to a 6"|
|Reviewer kHZc|8|-|-|
|Reviewer ZCVK|8|8|"Additional HELMET results and emerging skills analysis during RL have strengthen the contribution"|

And here is a summary of the additional experiments and analyses we conducted in response to the reviewers’ comments:

1. **Evaluation on more challenging long-context generation benchmarks.**
We additionally evaluated LoongRL on HELMET, a challenging long-context generation benchmark. The strong improvements demonstrate that the emergent plan–retrieve–reason–recheck pattern generalizes broadly across diverse long-context reasoning tasks, further validating the effectiveness of LoongRL.

2. **Additional ablations and clarifications.**
We included more ablation studies and clarified evaluation settings (e.g., seed variance and decoding setup) to ensure fair comparisons and to more clearly demonstrate the contribution of each component.

3. **Deeper analysis and discussion of KeyChain data design and emergent skills**.
In our responses to Reviewers 6KJK and WxW6, we clarified that the KeyChain data design is not UUID-specific. Any high-entropy, non-semantic key is sufficient, as this prevents lexical shortcuts and ensures explicit key–value retrieval. As clarified to Reviewer 6KJK, UUID-like strings were simply a convenient instantiation rather than a requirement. In our response to Reviewer ZCVK, we further analyzed what the model’s responses contain at different training stages and how LoongRL progressively acquires reasoning skills. This analysis reveals how the full plan–retrieve–reason–recheck pattern emerges during RL, and the reviewer acknowledged that these insights strengthened the contribution.

Finally, we sincerely thank all reviewers for their constructive questions and engagement. The additional experiments and analyses conducted during this discussion significantly strengthened our paper, and we will incorporate them into the revised version.

---

### Meta-Review · Area_Chair_dZD3 · 2025-12-26

**Summary:**

The authors introduced a data-driven RL framework for enhancing LLM long-context reasoning. The core contribution is a novel synthetic data generation technique that converts short QA into complex long-context tasks. The strength of this work is showing the generalizability of this approach to unseen longer contexts scenarios without full-length RL training. The extensive experimental results demonstrate the superiority of the proposed method.

**Reviewer Concerns:**

Initially, the reviewers raised some concerns on the experimental settings (e.g., most results are in-domain settings, only on UUID settings). Also, some reviewers provided some comments to check the performance on other benchmark datasets. But all of them are well resolved during the rebuttal periods.

**Reviewer Scores:**

Initially, the reviewers' scores are (6,4,8,8). After successful rebuttal, the reviewer's scores are changed to (6+, 6, 8, 8). 3 out of 4 reviewers are satisfied the responses and there is no remaining concerns that were raised by the reviewers.

---

### Decision · Program_Chairs · 2026-01-26

Accept (Oral)